# BRIDLE: Generalized Self-supervised Learning with Quantization

## Abstract

Self-supervised learning (SSL) has been a powerful approach for learning meaningful representations from unlabeled data across various domains, reducing the reliance on large labeled datasets. Inspired by BERT's success in capturing deep bidirectional contexts in natural language processing, similar frameworks have been adapted to other modalities such as audio, with models like BEATs extending the bidirectional training paradigm to audio signals using a single-codebook vector quantization (VQ) tokenizer to provide masked-prediction targets. The choice of *target tokenizer* in such bidirectional SSL frameworks is the central design decision that governs what the encoder is trained to predict; we argue that the single flat-VQ codebook is the binding constraint and that replacing it with a hierarchically-structured token space substantially improves downstream representation quality. To this end, we introduce BRIDLE (Bidirectional Residual Quantization Interleaved Discrete Learning Encoder), an improved *target tokenizer* for bidirectional SSL in the BEATs lineage that swaps the single-codebook VQ tokenizer for a hierarchical residual-quantization (RQ) tokenizer, while retaining the rest of the BEATs interleaved encoder–tokenizer training loop. RQ's additive multi-stage discretization provides a richer token geometry — up to $K_m^M$ effective combinations from $M$ codebooks of size $K_m$ — that the encoder learns to exploit, with the side effect of much higher post-training codebook usage than VQ. BRIDLE applies this tokenizer change across audio, image, and video. We evaluate it on audio classification benchmarks (state-of-the-art on AudioSet and ESC-50 in our reproduction setting), and show consistent improvements over the matched-total-codes VQ baseline on ImageNet-1K and Kinetics-400, with the largest gains observed under linear probing — evidence that the learned encoder representations are more linearly separable.

## 1 Introduction

By understanding the intrinsic structures within the data, *Self-supervised learning* (SSL) methods have reduced the reliance on large labeled datasets, which are often costly and time-consuming to produce. One of the most influential models in this space is BERT (Devlin et al., 2019), which has set new benchmarks in natural language processing tasks by capturing deep bidirectional contexts.

In the context of audio representation, capturing the intrinsic details of the audio input can be used in various downstream tasks (Niizumi et al., 2021). The introduction of transformer-based architectures and masked modeling approaches have revolutionized the field, with pioneering models such as SSAST (Gong et al., 2022a), MAE-AST (Baade et al., 2022), Audio-MAE (Huang et al., 2022). These models have demonstrated that discretizing continuous audio signals into discrete tokens can produce high-quality representations that are effective for a range of downstream tasks, including audio classification, speech recognition, and sound event detection (Chen et al., 2023; Baevski et al., 2020b; Hsu et al., 2021). BEATs (Chen et al., 2023) extends the bidirectional training paradigm to audio signals, achieving state-of-the-art results in various audio understanding tasks, using the idea of representing the inputs with *Vector Quantization* (VQ) and training the encoder to predict the mapped tokens in an unsupervised context.

However, despite their strengths, these frameworks face several challenges that limit their potential. Notably, their dependence on a single codebook for quantization may not capture the complex, multi-dimensional nature of audio signals, particularly in environments with diverse sound sources or varying acoustic conditions (van den Oord et al., 2017). Additionally, the training processes for the codebooks can be inefficient, as some code vectors remain unused or underutilized, leading to suboptimal representation learning and increased training time (Razavi et al., 2019). Several techniques have been introduced to optimize codebook utilization in VQ and *Residual VQ* (RVQ) frameworks (Kumar et al., 2023; Łańcucki et al., 2020), including $k$-means initialization for codebook vectors, randomized restarts for unused codebooks over several batches, and quantizer dropout to improve codebooks learning and usage.

*Residual Quantization* (RQ) has emerged as an effective method in various machine learning applications, particularly in the domain of computer vision and audio understanding, where it serves as a robust tool to reduce the dimensionality of data representations. Originally developed in the context of signal processing and vector quantization (Gray & Neuhoff, 1998; Gersho & Gray, 2012), RQ operates by recursively quantizing the residuals of previously quantized vectors, allowing it to capture fine-grained details that are missed by single-step quantization. This method has been successfully applied in large-scale image retrieval (Jegou et al., 2010) and approximate nearest neighbor search (Ge et al., 2013), demonstrating its efficiency in handling high-dimensional data. This hierarchical approach to quantization has proven to be highly beneficial in reducing information loss, making it an attractive option for tasks such as image compression and feature representation in vision models (Razavi et al., 2019; Lee et al., 2022).

ImageNet dataset Russakovsky et al. (2015) has been instrumental in advancing computer vision, serving as a benchmark for image classification tasks. Models pre-trained on ImageNet (Krizhevsky et al., 2012; Simonyan & Zisserman, 2015; He et al., 2016) have achieved remarkable performance and are widely used for transfer learning in various downstream tasks. Self-supervised learning methods on ImageNet have further improved the quality of learned representations without relying on labeled data. Contrastive learning frameworks such as SimCLR (Chen et al., 2020) and MoCo (He et al., 2020) maximize agreement between differently augmented views of the same image, capturing invariant features. Masked image modeling approaches such as MAE (He et al., 2022) apply masked modeling techniques to images, similar to BERT's masked language modeling.

Interestingly, audio understanding tasks can often be modeled similarly to vision models, as audio signals can be transformed into visual representations like mel-spectrograms. By converting raw audio data into mel-spectrograms, which encode frequency and temporal information as 2D matrices, the task of understanding audio becomes analogous to image processing. Residual quantization, when applied to image representations, helps to capture fine spectral details, much like it captures intricate visual features in image data. Tokenization and quantization methods have been utilized for efficient vector quantization in self-supervised learning frameworks such as those used in speech recognition and audio synthesis (van den Oord et al., 2017; Dhariwal et al., 2020). By quantizing the latent spaces of mel-spectrograms, models can efficiently capture important temporal and spectral patterns, crucial for audio classification and speech tasks (Baevski et al., 2020b; Chen et al., 2023; Chiu et al., 2022). This cross-domain applicability highlights residual quantization's versatility and its critical role in both vision and audio-based models, where it ensures efficient representation learning and high-quality reconstructions.

For video understanding, the Kinetics dataset (Kay et al., 2017) provides a large-scale collection of annotated video clips. Architectures employing 3D convolutions have benefited from pretraining on Kinetics, advancing action recognition and video analysis. Self-supervised approaches (Sun et al., 2019) and contrastive video representation learning methods (Qian et al., 2021) have extended bidirectional and self-supervised learning techniques to the temporal dimension, capturing complex spatiotemporal patterns.

Motivated by the quantization techniques in computer vision and audio compression, we introduce the Bidirectional Residual Quantization Interleaved Discrete Learning Encoder, or BRIDLE for short, which is built and improved based on BEATs (Chen et al., 2023), a bidirectional training process involves self-distilled training where the encoder acts as the teacher for the tokenizer, and the tokenizer trains the encoder to predict the tokens. In BRIDLE, we retain this interleaved encoder–tokenizer training and focus our changes

on the tokenizer, introducing a hierarchical residual-quantization target together with several improvements to codebook usage and training. Our contributions are:

1. We propose BRIDLE, a focused improvement to the *target tokenizer* of bidirectional SSL frameworks in the BEATs (Chen et al., 2023) lineage: we replace the single-codebook VQ tokenizer with a hierarchical residual-quantization (RQ) tokenizer, while retaining the rest of the BEATs encoder–tokenizer training loop. The RQ tokenizer provides an additive multi-stage discretization with up to $K_m^M$ effective token combinations from $M$ codebooks of size $K_m$ (e.g., $256^4$), in contrast to the $K_{\mathrm{VQ}} = MK_m$ flat alternatives of BEATs's single codebook (e.g., 1024) — a substantively richer token geometry that the encoder learns to exploit. We make the BEATs $\rightarrow$ BRIDLE delta concrete with a per-component comparison in Table 1.

2. We provide comprehensive evaluations of the framework in audio understanding through classification tasks on popular benchmarks, AudioSet (Gemmeke et al., 2017) and ESC-50 (Piczak, 2015), where we demonstrate state-of-the-art results. Additionally, we show competitive performance in image classification tasks with experiments on ImageNet-1K and in video classification with Kinetics-400. We also show consistent improvements in the encoder's downstream performance when using RQ compared to VQ.

3. We present a comprehensive analysis of codebook training within the BRIDLE training framework, covering aspects such as uniform weight initialization vs $k$-means, initial normalization of input embeddings, and resetting unused codes to prevent stagnation and encourage exploration.

4. On the theoretical side, we provide, to our knowledge, the first rigorous convergence analysis of the EMA codebook update with $\epsilon$-regularized normalization, showing almost-sure convergence of the cluster sizes $N_{i,t}$, the running embedding accumulators $\mathbf{m}_{i,t}$, and the codebook vectors $\mathbf{c}_{i,t}$ to explicit limits along the entire iterate sequence under mild assumptions on the latent and assignment trajectories (Appendix D).

## 2 Related Work

Self-supervised learning has become a cornerstone in representation learning across various data modalities, including audio, image, and video. In this section, we review the relevant literature in each modality, emphasizing residual quantization techniques, pretraining strategies, and self-supervised learning methods.

In computer vision, masked image modeling approaches, inspired by BERT, have been introduced with models such as BEiT (Bao et al., 2022; Peng et al., 2022; Wang et al., 2023) and MAE (He et al., 2022), where portions of the image are masked, and the model is trained to reconstruct the missing parts. These methods capture both local and global structures in images. Quantization techniques such as VQ-VAE (van den Oord et al., 2017) and its variants have been applied to image data, enabling discrete latent representations that facilitate powerful generative models. Residual quantization further enhances this by using multiple codebooks to capture complex image details, as seen in models such as RQ-VAE (Lee et al., 2022).

Several approaches have been proposed in the context of self-supervised learning for audio representation, with the utilization of contrastive learning, masked prediction, and quantization-based methods. Contrastive and clustering learning methods (Gong et al., 2022b; Chen et al., 2020; van den Oord et al., 2018; Seth et al., 2023) have been adapted to audio to learn discriminative representations (Saeed et al., 2021), which utilize contrastive loss functions to maximize similarity between positive pairs (augmented versions of the same audio clip) while minimizing similarity between negative pairs (different audio clips). These methods employ data augmentations specific to audio, such as time-shifting, pitch shifting, and noise addition, to enhance the robustness of learned features.

Recently, converting audio signals to discrete tokens processed by large language models (LLMs) has gained popularity due to the increasing demand for multimodal understanding. Llama 3 (Dubey et al., 2024) converts audios to tokens and the model could understand both text and audio seamlessly. Given the success of diffusion models on image generations, many works have explored audio generation using diffusions (Kreuk et al., 2023). However, modeling the raw audio waveform is prohibitively expansive for diffusion models, and different compact representations have been explored. AudioLM (Borsos et al., 2023) generates audio samples conditioned on text inputs, operating on discrete learned audio representations. SoundStream (Zeghidour et al., 2021) introduces an end-to-end neural audio codec that encodes audio into discrete tokens suitable

for downstream tasks. These approaches facilitate integration of audio data into LLMs, enabling advanced capabilities like audio-based question answering and generation.

Quantization-based techniques, particularly Vector Quantization (VQ), have been central to recent advances in self-supervised audio models. VQ techniques, as used in models such as VQ-Wav2Vec, map high-dimensional audio data into a finite set of discrete tokens or codes, facilitating robust speech representation learning (Baevski et al., 2020a). A larger codebook is needed to capture finer details of the audio signal. Nevertheless, scaling up code book in VQ suffers many challenges, especially skewed of codebook usage with a larger codebook. Instead, RQ, an extension of VQ, employs multiple codebooks in a hierarchical manner to capture finer details in audio data, improving the discretization of the latent space (Dhariwal et al., 2020).

On the other hand, video understanding poses unique challenges due to the additional temporal dimension. Pretraining frameworks have significantly advanced video representation learning through both supervised and self-supervised methods. Several models have been introduced to capture 3D information within videos, such as C3D (Tran et al., 2015), which uses 3D convolutions to learn spatiotemporal features. The two-stream architecture (Simonyan & Zisserman, 2014) processes spatial and temporal information separately using RGB frames and optical flow. I3D Carreira & Zisserman (2017) inflates 2D convolutional filters pre-trained on ImageNet into 3D, effectively transferring knowledge from images to videos.

Self-supervised learning has been adapted to video to leverage unlabeled data. Temporal order prediction methods such as Shuffle and Learn (Misra et al., 2016) and OPN (Lee et al., 2017) learn representations by predicting the correct temporal order of shuffled frames. Contrastive learning frameworks such as CVRL (Qian et al., 2021) extend contrastive methods to video by treating clips from the same video as positives. Masked video modeling approaches, such as VideoMAE (Tong et al., 2022), mask portions of the video input and train the model to reconstruct them, capturing spatiotemporal dependencies. While less explored in video, residual quantization techniques have potential for efficient video representation. Incorporating RVQ can improve the discretization of spatiotemporal data, facilitating tasks such as video compression and generative modeling.

BRIDLE represents a self-supervised learning framework that can work for all modalities, employing an encoder trained by a bidirectional pretraining process in which the encoder and tokenizer train each other in a self-distilled manner. BRIDLE leverages residual quantization to discretize continuous signals from image, audio and video data, and predicts tokens, akin to masked language models in NLP.

## 3   BRIDLE framework

The proposed BRIDLE framework focuses on integrating residual quantization Lee et al. (2022), improving codebook representation capability. To make BRIDLE's relationship to BEATs (Chen et al., 2023) concrete and self-contained for readers who are not BEATs-fluent: BRIDLE *retains* BEATs's interleaved encoder–tokenizer training loop, masked-prediction encoder loss, EMA-style codebook update, and ViT-base architecture; what we *change* is the tokenizer's discretization mechanism (single-codebook VQ $\rightarrow$ hierarchical residual quantization) together with the codebook training techniques (initialization, code resetting, normalization). Table 1 summarizes the differences cell by cell. We position BRIDLE as a focused study of the tokenizer mechanism within an existing bidirectional SSL framework, rather than as a wholly new framework.

We emphasize that the matched-total-codes equality ($M \cdot K_m = 4 \cdot 256 = 1024$) keeps the *number of atoms* comparable between BEATs (VQ) and BRIDLE (RQ), but the two setups span fundamentally different token spaces: VQ's tokens lie in a finite set of size $K$, while RQ's additive composition can in principle produce up to $K^M$ distinct tokens organized along $M$ residual stages. In addition to making evaluation easier, this richer geometry is the structural change we expect to drive the downstream gains; we revisit this in the analysis of Section 5 and Appendix G. We do *not* claim that the gains stem purely from greater code utilization in the BEATs sense; rather, the hierarchical-residual structure provides a more expressive token space that the encoder learns to exploit, with utilization being one downstream observable (see Appendix G, Table 12).

BRIDLE comprises four main components:

Table 1: Concrete differences between BEATs and BRIDLE. Cells marked "same" carry the BEATs (Chen et al., 2023) setting unchanged.

| Component | BEATs (Chen et al., 2023) | BRIDLE (this work) |
|---|---|---|
| Encoder/decoder architecture | ViT-base | same |
| Encoder–tokenizer schedule | Interleaved, two iterations | same |
| Encoder loss | Masked discrete-label prediction | same |
| Tokenizer estimator alignment | Cosine similarity to encoder $\mathbf{z}$ | same |
| EMA codebook update | Yes | same (with $\epsilon$-regularized normalization) |
| Discretization | VQ, $M = 1$ codebook of $K = 1024$ codes | RQ, $M = 4$ codebooks of $K_m = 256$ codes |
| Token computation | Pick one code: $\mathbf{q} = \mathbf{c}_{i^*}$ | Sum residual codes: $\mathbf{q} = \sum_{m=1}^{M} \mathbf{c}_{i^*(m)}^{(m)}$ |
| Effective token combinations | $K = 1024$ | up to $K_m^M = 256^4 \approx 4{\times}10^9$ |
| Codebook initialization | Random | Uniform on $[-1, 1]^D$ or $k$-means (Sec. 3) |
| Unused-code reset | Not used | Reset on first batch if $N_i < T_u$ |

**Main Encoder** $E(\cdot; \theta_E)$: Maps input features $\mathbf{x} \in \mathbb{R}^{T \times F}$, where $T$ is the time (or, for image and video, the spatio-temporal token) dimension and $F$ is the feature dimension, to a latent representation $\mathbf{z} = E(\mathbf{x}; \theta_E) \in \mathbb{R}^{T \times D}$, where $D$ is the dimension of the latent space. Throughout the paper, we reserve the symbol $\mathbf{z}$ exclusively for the main encoder's output.

**Tokenizer** $T(\cdot; \theta_T)$: Comprises two sub-components that together produce discrete tokens: (i) a *tokenizer encoder* $E_T(\cdot; \theta_T^{\text{enc}})$ — an independent network with the same architecture as $E$ — that maps $\mathbf{x}$ to its own latent representation $\tilde{\mathbf{z}} = E_T(\mathbf{x}; \theta_T^{\text{enc}}) \in \mathbb{R}^{T \times D}$; and (ii) a residual-quantization module operating on the codebooks $\mathcal{C}_{1:M}$ that discretizes $\tilde{\mathbf{z}}$ into a sequence of tokens $\mathbf{q} = Q(\tilde{\mathbf{z}}; \mathcal{C}_{1:M}) \in \mathbb{Z}^{T \times M}$, where $M$ is the number of codebooks. We use the tilde to distinguish the tokenizer-encoder latent $\tilde{\mathbf{z}}$ from the main-encoder latent $\mathbf{z}$; this distinction is essential for the codebook-loss definition in (5).

**Main Decoder** $D(\cdot; \theta_D)$: Predicts the tokens output by the tokenizer, facilitating the reconstruction of the input.

**Tokenizer Estimator** $\text{TE}(\cdot; \theta_{\text{TE}})$: Acts on the quantized tokenizer output $\mathbf{q}$ and predicts the main encoder's embeddings $\mathbf{z}$, ensuring alignment between the (frozen) main encoder and the (trainable) tokenizer during the tokenizer training phase.

We adopt the interleaved training framework from BEATs (Chen et al., 2023), which consists of the encoder training phase, and the tokenizer training phase. When we train the encoder, the main encoder and decoder learn to predict masked labels produced by the tokenizer. On the other hand, when the tokenizer is trained, the tokenizer encoder, codebooks, and the estimator learn the embeddings produced by their teacher, the main encoder. Figure 1 details the framework training process. The framework begins with the encoder training phase, utilizing a cold start tokenizer for the first iteration. In subsequent iterations, the tokenizer is reset and trained during a dedicated tokenizer training phase, each followed by an encoder training phase. This approach establishes an interleaved training framework. We evaluate the main encoders in downstream tasks after every encoder training phase for all modes.

Algorithm 1 states the full procedure explicitly and annotates each step as either inherited unchanged from BEATs ([**B**]) or specific to BRIDLE ([**R**]). We make the inherited-vs-new split deliberately visible in response to the natural question of whether the interleaving is a genuinely new procedure or an adaptation: *the interleaved encoder–tokenizer schedule, the cold-start first iteration, the tokenizer reset between iterations, the masked-prediction encoder loss, and the EMA codebook update are all inherited from BEATs* (lines marked [**B**]); *the residual-quantization target computation, the k-means codebook initialization, and the first-batch unused-code reset are the BRIDLE changes* (lines marked [**R**]). In other words, BRIDLE does not propose

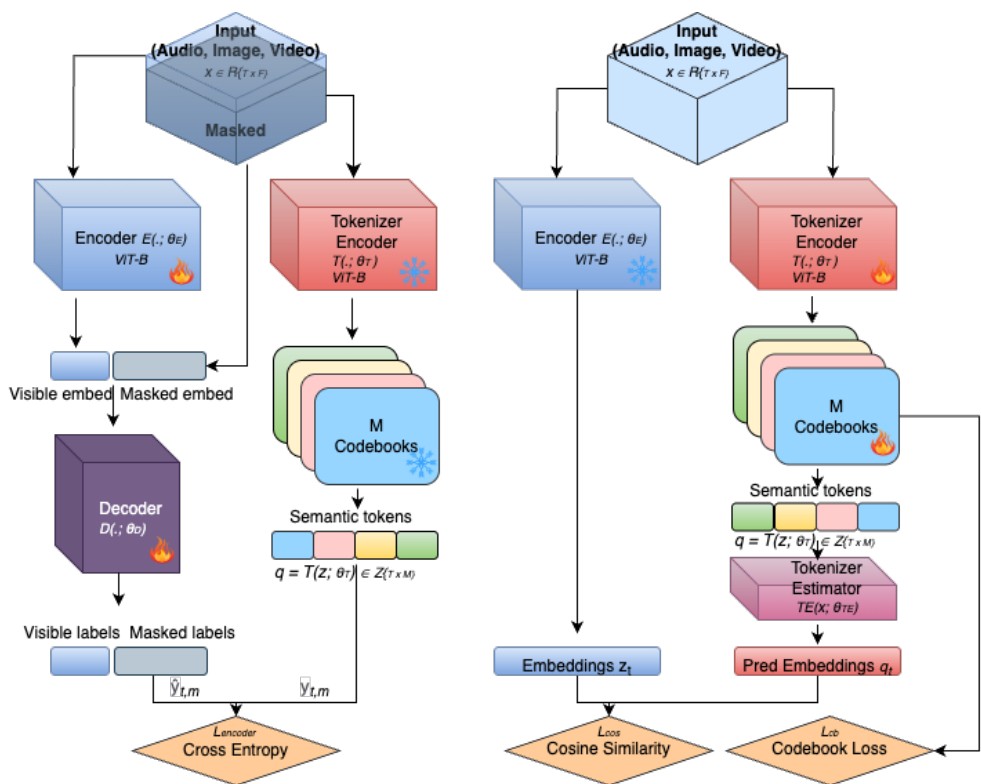

Figure 1: BRIDLE training framework, which contains an encoder training phase (left), and a tokenizer training phase (right). The main encoder $E$ produces the latent $\mathbf{z}$; the tokenizer encoder $E_T$ produces its own latent $\tilde{\mathbf{z}}$, which is quantized via the codebooks $\mathcal{C}_{1:M}$ into tokens $\mathbf{q}$. During the encoder training phase, the tokenizer (and thus $\tilde{\mathbf{z}}$) is frozen and provides target tokens; during the tokenizer training phase, the main encoder (and thus $\mathbf{z}$) is frozen and provides the alignment target for the tokenizer estimator.

a new training loop; it substitutes a hierarchical-residual tokenizer, and its associated codebook-training techniques, into an existing loop. This is exactly the "focused tokenizer-mechanism improvement, not a new framework" positioning of Table 1, now made concrete at the algorithmic level.

## 3.1 Residual Quantization

To enhance the discretization process, we employ residual quantization in BRIDLE. RQ uses a hierarchy of codebooks $\{\mathcal{C}_1, \mathcal{C}_2, \ldots, \mathcal{C}_M\}$, where each codebook $\mathcal{C}_m \in \mathbb{R}^{K_m \times D}$ contains $K_m$ code vectors of dimension $D$. The quantization process is performed in multiple stages, such that every codebook quantizes the residual error from the previous stage (Razavi et al., 2019).

Given a tokenizer-encoder latent vector $\tilde{\mathbf{z}}_t \in \mathbb{R}^D$ at position $t$, the quantization at each stage $m$ is defined as follows. Initially, the residual is set to be $\mathbf{e}_1 = \tilde{\mathbf{z}}_t$. At each stage $m$, we select the code vector $\mathbf{c}_{i^*(m)}^{(m)} \in \mathcal{C}_m$ that minimizes the distance to the residual error $\mathbf{e}_m$, for $\mathbf{c}_i^{(m)} \in \mathcal{C}_m$:

$$i^*(m) = \arg\min_{i=1,2,\ldots,K_m} \left\| \mathbf{e}_m - \mathbf{c}_i^{(m)} \right\|^2. \tag{1}$$

We then update the residual for the next stage:

$$\mathbf{e}_{m+1} = \mathbf{e}_m - \mathbf{c}_{i^*(m)}^{(m)}. \tag{2}$$

---

**Algorithm 1** BRIDLE interleaved pretraining. Each step is tagged **[B]** if inherited unchanged from BEATs (Chen et al., 2023) or **[R]** if it is a BRIDLE change. RQ denotes the $M$-stage residual assignment of Section 3.1 (returns both the summed embedding $\mathbf{q}$ and the per-stage code indices $\mathbf{y} \in \mathbb{Z}^{T \times M}$).

---

**Require:** unlabeled data $\mathcal{X}$; bidirectional iterations $R=2$; mask ratio $\rho$; RQ depth $M$, per-codebook size $K_m$; reset threshold $T_u$

 1: initialize main encoder $E$ and decoder $D$       **[B]**
 2: set tokenizer encoder $E_T$ to a random linear projection (cold start)       **[B]**
 3: **for** iteration $r = 1, \ldots, R$ **do**
 4:    **if** $r > 1$ **then**       ▷ tokenizer training phase; $E$ frozen as teacher **[B]**
 5:       reset $E_T$, tokenizer estimator TE, and codebooks $\mathcal{C}_{1:M}$       **[B]**
 6:       initialize $\mathcal{C}_{1:M}$ by $k$-means (default) or uniform on first batch of $\tilde{\mathbf{z}}$       **[R]**
 7:       on the first batch, reset any code with usage $N_i < T_u$       **[R]**
 8:       **for** each batch $\mathbf{x} \in \mathcal{X}$ **do**
 9:          $\tilde{\mathbf{z}} \leftarrow E_T(\mathbf{x}); \quad \mathbf{z} \leftarrow \mathrm{sg}[E(\mathbf{x})]$       **[B]**
10:          $(\mathbf{q}, \cdot) \leftarrow \mathrm{RQ}(\tilde{\mathbf{z}}, \mathcal{C}_{1:M})$       **[R]**
11:          update $E_T$, TE on $\mathcal{L}_{\text{tokenizer}} = \mathcal{L}_{\text{cb}} + \lambda_{\cos} \mathcal{L}_{\cos}$; EMA-update $\mathcal{C}_{1:M}$       **[B]/[R]**
12:       **end for**
13:    **end if**
14:    *encoder training phase; tokenizer $T=(E_T, \mathcal{C}_{1:M})$ frozen, gives targets*       **[B]**
15:    **for** each batch $\mathbf{x} \in \mathcal{X}$ **do**
16:       $(\cdot, \mathbf{y}) \leftarrow \mathrm{RQ}(E_T(\mathbf{x}), \mathcal{C}_{1:M})$       ▷ single flat index if VQ **[R]**
17:       mask $\mathbf{x}$ at ratio $\rho$; predict $\hat{\mathbf{y}}$ with $E, D$       **[B]**
18:       update $E, D$ on masked-prediction loss $\mathcal{L}_{\text{encoder}}$ of Eq. (4)       **[B]**
19:    **end for**
20:    evaluate $E$ on downstream tasks       **[B]**
21: **end for**

---

After $M$ stages, the final quantized representation $\mathbf{q}_t$ is obtained by summing the selected code vectors from all stages:

$$\mathbf{q}_t = \sum_{m=1}^{M} \mathbf{c}_{i^*(m)}^{(m)}. \tag{3}$$

### 3.2   Loss Functions

The training of the encoder and tokenizer involves two different set of loss functions. The *encoder loss* is a masked cross-entropy loss that measures the discrepancy between the tokens predicted by the main decoder and the actual tokens output by the tokenizer. This loss ensures that the encoder learns to produce representations that the decoder can accurately translate back into the original token sequence:

$$\mathcal{L}_{\text{encoder}} = -\frac{1}{T} \sum_{t=1}^{T} \sum_{m=1}^{M} \mathbf{y}_{t,m} \log \hat{\mathbf{y}}_{t,m}, \tag{4}$$

where $\mathbf{y}_{t,m}$ is the ground truth token from the tokenizer at time step $t$ and codebook $m$, $\hat{\mathbf{y}}_{t,m}$ is the predicted probability distribution over the token vocabulary by the main decoder at time step $t$ and codebook $m$. The *tokenizer loss* comprises two components:

**Codebook Loss ($\mathcal{L}_{\text{cb}}$):** Encourages effective and accurate mapping and representation of the tokenizer encoder's latent vectors by penalizing the distance between $\tilde{\mathbf{z}}_t$ and its quantization $\mathbf{q}_t$. It includes a commitment term to ensure that the tokenizer encoder output stays close to the code vectors:

$$\mathcal{L}_{\text{cb}} = \frac{1}{T} \sum_{t=1}^{T} \left( \|\mathrm{sg}[\tilde{\mathbf{z}}_t] - \mathbf{q}_t\|^2 + \beta \|\tilde{\mathbf{z}}_t - \mathrm{sg}[\mathbf{q}_t]\|^2 \right), \tag{5}$$

where sg[·] denotes the stop-gradient operator, which is the identity at the forward pass and has zero partial derivatives (van den Oord et al., 2017), and $\beta$ is a hyperparameter controlling the strength of the commitment term. The two terms in (5) play complementary roles, made decoupled by the asymmetric stop-gradient pattern. The first term $\|\text{sg}[\tilde{\mathbf{z}}_t] - \mathbf{q}_t\|^2$ updates the *codebooks* so that they track the empirical distribution of tokenizer-encoder features; this is what makes the codebooks data-adapted. The second term $\beta\|\tilde{\mathbf{z}}_t - \text{sg}[\mathbf{q}_t]\|^2$, the standard *commitment* term (van den Oord et al., 2017), updates the *tokenizer encoder* $E_T$ so that $\tilde{\mathbf{z}}_t$ remains close to the codebook entry it is being snapped to; without it, $E_T$ may drift relative to the (slowly-updated) codebooks and quantization error can compound, destabilizing training. We highlight that gradients from (5) do not flow into the main encoder $E$ — neither term involves $\mathbf{z}_t$ — preserving the role of the main encoder as a teacher during the tokenizer training phase.

**Cosine Similarity Loss ($\mathcal{L}_{\cos}$):** Measures the cosine similarity between the tokenizer estimator's output and the main encoder's embeddings, promoting alignment between the two components:

$$\mathcal{L}_{\cos} = 1 - \frac{\sum_{t=1}^{T} \text{TE}(\mathbf{q}_t) \cdot \mathbf{z}_t}{\sum_{t=1}^{T} \|\text{TE}(\mathbf{q}_t)\|\|\mathbf{z}_t\|}. \tag{6}$$

This loss component ensures the tokenizer captures proper latent information given by its teacher, the main encoder, through the process of predicting the target embeddings $\mathbf{z}_t$ from the discrete tokens $\mathbf{q}_t$. The total tokenizer loss combines (5) and (6):

$$\mathcal{L}_{\text{tokenizer}} = \mathcal{L}_{\text{cb}} + \lambda_{\cos}\mathcal{L}_{\cos}, \tag{7}$$

where $\lambda_{\cos}$ is a weighting factor balancing the two loss components. The two terms target complementary failure modes. $\mathcal{L}_{\text{cb}}$ controls *quantization fidelity* in the tokenizer's own representation space — an intrinsic property of the tokenizer. $\mathcal{L}_{\cos}$ controls *cross-component alignment* so that the tokens chosen by the tokenizer remain informative for the masked-prediction objective the main encoder is trained against. Without $\mathcal{L}_{\cos}$, the tokenizer can learn a high-fidelity discretization that is nevertheless decoupled from the main encoder's representation; without $\mathcal{L}_{\text{cb}}$, alignment can be achieved by collapsing the codebooks. The weighting factor $\lambda_{\cos}$ trades off these two failure modes, with the value used in our experiments reported in Appendix A.

### 3.3 Codebook Training Techniques

To optimize codebook training, we adopt the following key techniques within the BRIDLE framework:

**Exponential Moving Average (EMA) update** (van den Oord et al., 2017): The EMA update has been shown to be effective in codebook learning (Chen et al., 2023; Lee et al., 2022). The EMA provides smoother updates and reduces variance, leading to more stable convergence. In Appendix D, we show the convergence of the EMA update for all potential codebooks, given minor assumptions on the convergence of latent vectors and sufficiently large number of iterations. To our knowledge, this is a novel theoretical contribution.

**Resetting Unused Codes** (Kumar et al., 2023): To improve codebook mapping, we reset unused or underutilized codes in the codebook $\mathcal{C}_m$. If the usage count $N_i$ for a code vector $\mathbf{c}_i$ falls below a threshold $T_u$, we reinitialize $\mathbf{c}_i$ based on the current data distribution:

$$\text{Rand}\{\tilde{\mathbf{z}}_t\} \to \mathbf{c}_i \quad \text{if} \quad N_i < T_u,$$

where $\{\tilde{\mathbf{z}}_t\}$ are the tokenizer-encoder latent vectors feeding the codebook. This prevents codebook collapse and encourages diverse representation learning. In our implementation, resetting unused codes is applied only in the first batch of the tokenizer pretraining phase, so that it does not interfere with codebook training.

**Code Embedding Initialization - Uniform or $k$-means:** At the start of training, we initialize codebook entries by one of two strategies. *Uniform initialization*: since the latent vectors $\{\tilde{\mathbf{z}}_t\}$ feeding the codebook are pre-normalized so that each coordinate has approximately unit variance and lies in $[-1, 1]$, we draw each codebook entry $\mathbf{c}_i \in \mathbb{R}^D$ as i.i.d. uniform on $[-1, 1]^D$ (i.e., uniform per coordinate, matched to the typical range of $\tilde{\mathbf{z}}_t$). The intent is three-fold: (i) bound entries away from zero, (ii) span the typical range of the latent so all entries have non-negligible probability of being selected on the first batch, and (iii) avoid biasing entries

toward any pre-clustered structure (which the $k$-means alternative does explicitly). *k-means initialization*: run $k$-means on the first batch of latent vectors $\{\tilde{\mathbf{z}}_t\}$ for 10 iterations and use the resulting cluster centers as initial codebook entries. Empirically (Tables 7–8 and 9–10), $k$-means initialization is consistently more stable and produces equal or better downstream performance, so we recommend it as the default and report uniform-init only as a sensitivity baseline.

# 4 Experiments and Results

We evaluate the BRIDLE model across three modalities: audio, image, and video. For each modality, we perform pretraining on large-scale datasets and fine-tuning on standard benchmarks to assess the model's performance comprehensively. Our goal is to demonstrate that integrating residual quantization, improving codebook training leads to performance improvement in self-supervised representation learning. For important audio benchmark on AudioSet (Gemmeke et al., 2017), and image benchmark on ImageNet-1K (Russakovsky et al., 2015), we also provide linear probing performances. These results demonstrate more noticeable improvements, especially where the model's fine-tuning results are relatively saturated, given that we are using the base ViT architecture for all modes. For audio, we experiment our framework on the *AudioSet* dataset (Gemmeke et al., 2017), which contains over 2 million 10-second sound clips drawn from YouTube videos and labeled across 527 classes, and the *ESC-50* dataset (Piczak, 2015), a collection of 2,000 5-second environmental audio recordings across 50 classes. For image data, we use the popular *ImageNet-1K* dataset (Deng et al., 2009), which consists of approximately 1.28 million training images and 50,000 validation images in 1,000 classes. For video data, we employ the *Kinetics-400* dataset (Kay et al., 2017), which contains around 240,000 video clips annotated with 400 human action classes.

## 4.1 Model Architecture

We use the base ViT architectures for all modes. The tokenizer encoder has the same architecture as the main encoder. Similar to BEATs (Chen et al., 2023) and inspired by the success of random linear projection tokenizer in the literature (Chiu et al., 2022; Dubey et al., 2024), for iteration 1, we use a simple linear projection as the cold start tokenizer encoder. The BRIDLE framework incorporates residual quantization with $M = 4$ codebooks for each modality. The choice $M = 4$ was selected from the audio-iter1 ablation in Table 7, which compares 4 sets of 256 codes (default) against 3 sets of 256 codes ($-0.4$ FT mAP, $-0.86$ LP mAP) and other variants; the $M = 4$ setting was the empirical sweet spot among the configurations we tested and is consistent with the $M = 4$ stages typical of RQ-VAE-style tokenizers in the vision literature (Lee et al., 2022). We did not observe further gains from $M > 4$ in early experimentation under our matched-total-codes budget. We test this architecture against the VQ version of the model, where we use $M = 1$ codebook with $K = 1024$ codes, keeping the total number of atoms matched ($M \cdot K_m = 1024 = K_{\text{VQ}}$) so that the comparison isolates the effect of *how* the atoms are organized (flat vs hierarchical residual) rather than *how many* atoms are available; we discuss the limits of this matching in Section 3 (Table 1). For RQ, each latent vector is mapped to $M = 4$ codes whose sum yields the quantized representation, providing the richer hierarchically-structured token geometry analyzed in Section 5. We refer to Appendix A for hyperparameters details, Appendix C to support hyperparameters choices and Appendix E for model configurations.

## 4.2 Training Procedure

We first pretrain the models for two bidirectional iterations, then evaluate the models on corresponding classification tasks on the datasets mentioned above. Due to the empirical observations from BEATs, the performance of the model shows negligible improvements with additional training iterations. We observe similar trend in our experiments, hence we stop at 2 iterations for all modes. During the encoder training phase of audio and image, we use masking ratio of 0.8, and for the tokenizer training phase, we do not apply masking. For the video modality, the process is similar to audio and image training, except for the usage of tube masking with 0.9 ratio to increase reconstruction difficulty and prevent information leakage (Tong et al., 2022).

**Audio**: The model is pretrained on the AudioSet-2M dataset for 130 epochs per encoder training phase, and 30 epochs for tokenizer training. We then evaluate the model with fine-tuning and linear probing performances on AudioSet-2M and AudioSet-20K, as well as fine-tuning performance on ESC-50.

**Image**: The model is pretrained on ImageNet-1K following the procedure in AudioMAE (Huang et al., 2022). We use 400 encoder training epochs and 100 tokenizer training epochs. Standard data augmentation techniques such as random cropping and horizontal flipping are applied. After pretraining, we fine-tune the model on the ImageNet-1K training set and evaluate it on the validation set.

**Video**: We pretrain the model on the Kinetics-400 training set for 800 epochs per encoder training phase, and 200 epochs for tokenizer training phase, following the setup from VideoMAE (Tong et al., 2022). During training, video clips are sampled with temporal strides to capture motion dynamics effectively. Data augmentations include random cropping and horizontal flipping in both spatial and temporal dimensions. Hence, our video models behave like multi-frame image models. After pretraining, the model is fine-tuned on the Kinetics-400 training set and evaluated on the validation set.

### 4.3  Results

We evaluate the model using mean Average Precision (mAP) for multi-label classification tasks on AudioSet, and classification accuracy with 5-fold cross validation for ESC-50, which is a multi-class single-label dataset. We report Top-1 and Top-5 classification accuracies on the ImageNet-1K validation set. For Kinetics-400, we evaluate the model using Top-1 and Top-5 classification accuracies on the validation set. We present the audio classification results on the AS-2M, AS-20K, and ESC-50 datasets in Table 2. We compare different models using both fine-tuning and linear probing evaluations. Table 2 demonstrates that incorporating *residual quantization* (RQ) into the BEATs framework enhances performance across all evaluated datasets and metrics. The reproduction of BEATs (Chen et al., 2023) on ∼15% missing data from AudioSet-2M shows a minor drop of ∼0.5% in fine-tuning mAP. We emphasize that all rows of Table 2 (both the BEATs-reproduction baseline and the BRIDLE variants) are trained and evaluated on the *identical* reduced AudioSet-2M subset, so the within-table comparisons between BEATs and BRIDLE are internally controlled — the 0.5% drop is the BEATs-paper-vs-our-reproduction gap, not a BEATs-vs-BRIDLE gap. The missingness arises from YouTube-side video unavailability rather than content-conditional sampling, and Appendix B provides per-segment availability statistics supporting that this loss is approximately uniform across classes. Nevertheless, BRIDLE utilizing RQ codebooks and $k$-means initialization achieves the best results across all evaluation tasks. We have shown improvements compared to the benchmark of BEATs reproduction, which provided state-of-the-art results in the context of audio self-supervised training.

Table 2: Audio classification results on the AS-2M, AS-20K, and ESC-50 datasets. Fine-tuning (FT) and linear probing (LP) results are reported in mAP or accuracy (%).

| Model | AS-2M mAP | | AS-20K mAP | | ESC-50 FT | |
|---|---|---|---|---|---|---|
| | FT | LP | FT | LP | mAP | Acc |
| (VQ) BEATs$_{iter1}$ Chen et al. (2023) | 46.60 | 26.76 | 35.36 | 23.32 | 97.19 | 93.75 |
| (VQ) BEATs$_{iter2}$ Chen et al. (2023) | 47.64 | 27.04 | 36.89 | 23.41 | 97.86 | 94.55 |
| (uniform) BRIDLE$_{iter1}$ | 47.19 | 29.15 | 36.99 | 25.54 | 97.47 | 94.45 |
| (uniform) BRIDLE$_{iter2}$ | 47.56 | 28.71 | 37.72 | 25.51 | 97.83 | 94.85 |
| (k-means) BRIDLE$_{iter1}$ | 47.17 | **29.58** | 37.03 | 26.07 | 97.61 | 94.45 |
| (k-means) BRIDLE$_{iter2}$ | **47.99** | 29.56 | **38.08** | **26.37** | **97.89** | **94.95** |

In the vision context, the image classification results on the ImageNet-1K dataset are presented in Table 3, where we report both fine-tuning and linear probing accuracies. The fine-tuning action recognition results on the Kinetics-400 dataset are presented in Table 4. We report fine-tuning Top-1 and Top-5 accuracies.

Table 3 and Table 4 highlight the effectiveness of residual quantization in enhancing image and video representation learning, leading to better performance in action recognition tasks. Across all three modalities—audio, image, and video, the incorporation of residual quantization into the bidirectional pretraining framework consistently improves performance. The benefits are more pronounced in linear probing evaluations, suggesting

Table 3: Image classification results on the ImageNet-1K dataset. Fine-tuning (FT) results are reported in Top-1 and Top-5 accuracies (%); linear probing (LP) results follow the same convention. The upper block reports representative external self-supervised baselines at comparable model scale (numbers from the original references); the lower blocks compare VQ vs RQ ($k$-means) within BRIDLE. Bold denotes the best result *within the BRIDLE block*.

| Model | FT Acc (%) | | LP Acc (%) | |
|---|---|---|---|---|
| | Top-1 | Top-5 | Top-1 | Top-5 |
| ViT384-B-JFT300M (Dosovitskiy et al., 2021) | 79.9 | — | — | — |
| DINO (Caron et al., 2021) | 82.8 | — | — | — |
| BEiT (Bao et al., 2022) | 83.2 | — | — | — |
| MoCo v3 (Chen et al., 2021) | 83.2 | — | — | — |
| MAE (He et al., 2022) | 83.6 | — | — | — |
| (VQ) BRIDLE$_{iter1}$ | 79.10 | 93.86 | 44.82 | 69.22 |
| (VQ) BRIDLE$_{iter2}$ | 80.26 | 94.41 | 53.21 | 76.89 |
| ($k$-means) BRIDLE$_{iter1}$ | 80.00 | 94.30 | 52.96 | 76.60 |
| ($k$-means) BRIDLE$_{iter2}$ | **81.10** | **95.00** | **56.30** | **79.24** |

Table 4: Action recognition results on the Kinetics-400 dataset. Fine-tuning results are reported in Top-1 and Top-5 accuracies (%). The upper block lists representative external self-supervised baselines at the ViT-Base scale (numbers from the original references; — indicates not reported by the cited work); the lower blocks compare VQ vs RQ ($k$-means) within BRIDLE. Bold denotes the best result *within the BRIDLE block*. Note that BRIDLE on video uses the *same* encoder–tokenizer recipe as on audio and image with no modality-specific tuning, so the comparison to video-specialized baselines is informative but not the headline contribution; the headline is the consistent VQ→RQ improvement across modalities.

| Model | Top-1 Acc (%) | Top-5 Acc (%) |
|---|---|---|
| VideoMAE (Tong et al., 2022) | 80.0 | — |
| (VQ) BRIDLE$_{iter1}$ | 68.69 | 87.44 |
| (VQ) BRIDLE$_{iter2}$ | 71.32 | 88.86 |
| ($k$-means) BRIDLE$_{iter1}$ | 71.36 | 89.03 |
| ($k$-means) BRIDLE$_{iter2}$ | **72.90** | **90.08** |

that RQ enables the encoder to learn more generalizable and robust representations. The use of $k$-means initialization for codebooks further enhances performance, indicating that careful codebook initialization is crucial for optimizing quantization-based models.

Relative to the external baselines reported in Tables 3 and 4, BRIDLE on ImageNet-1K is competitive with — though slightly below — masked-image-modeling methods such as MAE and BEiT, while reaching its score after only two bidirectional iterations (each markedly shorter than the 800–1600 epoch schedules typical of MAE or BEiT) and natively producing a discrete-token output suitable for downstream generative or retrieval models; see Appendix F for a more detailed positioning. On Kinetics-400, BRIDLE trails the modality-specialized VideoMAE in absolute accuracy: this is expected, as BRIDLE applies the unified audio/image/video recipe with no video-specific architectural changes, and the cross-modality generality is the contribution we emphasize. The within-modality VQ→RQ improvement (the second-vs-third blocks of each table) is the controlled comparison that supports the paper's main claim, and that improvement is consistent across all three modalities.

**A note on variance.** Tables 2–4 report single-run numbers per cell. Multi-seed pretraining on AS-2M, ImageNet-1K, and Kinetics-400 at our schedules (e.g., $130 + 30$ epochs $\times$ 8 nodes $\times$ 8 GPUs for AS-2M alone) is computationally substantial; we did not perform multi-seed runs at this revision. The one setting where the protocol itself produces multiple measurements is ESC-50, where reported accuracy is the mean over 5-fold cross-validation. Within-table differences of magnitude comparable to plausible single-run noise on any one benchmark should therefore be interpreted with caution; what we believe is the more robust signal is the

*consistent direction* of the VQ→RQ improvement across all three modalities and across both fine-tuning and linear probing — eight directional comparisons in Tables 2–4, all favoring RQ. Uniform initialization is, as the reviewer notes, the more fragile of the two initialization strategies (Tables 7–8), which is part of why we recommend $k$-means as the default.

## 5    Empirical Discussions

**Geometry vs. utilization: what does RQ actually buy?** A natural question — raised explicitly in our review process — is whether the gains BRIDLE achieves over the matched-total-codes BEATs (VQ) baseline come from (a) higher *utilization* of the codebook entries, or (b) a richer *geometry* of the token space. Recent literature on simplicity bias (Tishby et al., 2000; Valle-Perez et al., 2018; Wilson, 2025) argues that learning generally biases neural networks toward simpler representations, so codebooks are expected to be under-utilized unless the size of the codebook is calibrated to the genuine capacity needed by the data; under that view, raw utilization is not, in itself, the right thing to optimize.

We agree with this framing and clarify the BRIDLE contribution accordingly. Empirically, the single VQ codebook in our setting collapses to a Code Usage Rate (CUR) of $\sim 0.21$ post-training (Appendix G, Table 12, BRIDLE$_{\text{iter2}}$ VQ row), meaning the trained tokenizer effectively uses only $\sim 215$ of 1024 atoms — close to the size of just one of RQ's $K_m = 256$ codebooks. This is consistent with the simplicity-bias prediction: the network found a "simple-enough" representation in the flat 1024-atom space and stayed there. RQ's hierarchical-residual codebooks, by contrast, retain CUR $\approx 1.00$ across all four codebooks. Because the matched-total-codes equality $M \cdot K_m = K_{\text{VQ}}$ does *not* match the effective token spaces — VQ provides $K_{\text{VQ}} = 1024$ alternatives per token, while RQ's additive composition can in principle yield up to $K_m^M = 256^4 \approx 4 \times 10^9$ distinct tokens organized along $M$ residual stages — what RQ provides is best understood as a *better representation geometry* for the tokenizer rather than a fix to a utilization defect. Utilization is then a downstream observable of that better geometry: the encoder learns to make use of structurally distinct tokens rather than collapsing to a small subset of a flat alternative space. We therefore frame BRIDLE's primary mechanism as "richer hierarchically-structured token geometry," with the catastrophic VQ utilization collapse documented as evidence that flat-VQ targets do not provide enough useful structure in this setting, not as the headline contribution.

**Attributing the gains: residual quantization vs. $k$-means initialization.** Because the two levers BRIDLE introduces — the residual quantizer and the $k$-means codebook initialization — could in principle be confounded, we isolate them directly from the audio results in Table 2, which reports all three relevant configurations along a single axis: single-codebook VQ (BEATs), RQ with uniform (data-agnostic) initialization, and RQ with $k$-means initialization. Reading down the AS-2M linear-probing column at iteration 2, the total VQ→RQ($k$-means) improvement of +2.52 mAP (27.04 → 29.56) decomposes additively into a *quantizer* effect and an *initialization* effect: holding initialization data-agnostic, replacing VQ with RQ moves LP mAP from 27.04 to 28.71 (+1.67); then, holding the quantizer fixed at RQ, switching uniform to $k$-means initialization moves it from 28.71 to 29.56 (+0.85). The same ordering holds at iteration 1 (+2.39 from the quantizer, +0.43 from $k$-means; total +2.82). Two conclusions follow that speak directly to the attribution concern. First, *the residual quantizer, not the initialization, is the dominant lever* — roughly two-thirds to five-sixths of the linear-probing gain is attributable to VQ→RQ with initialization held fixed, and $k$-means contributes a smaller, consistently positive increment. Second, because the VQ→RQ effect is present at *both* bidirectional iterations, it is not an artifact of the training schedule (the iteration axis); the schedule improves all configurations in parallel rather than manufacturing the RQ advantage. Fine-tuning mAP is comparatively saturated at the base-ViT scale (Section 4.3), which is why this decomposition is cleanest in linear probing, the setting the reviewers correctly identify as the more representation-diagnostic metric. We note one honest limitation of this isolation: the clean three-configuration comparison (VQ / RQ-uniform / RQ-$k$-means) is available for audio, whereas Tables 3–4 report only VQ vs. RQ($k$-means) for image and video, so for those modalities the quantizer and initialization effects are not separately identified — we therefore restrict the quantitative attribution claim to audio and treat the image/video results as confirming only the *combined* VQ→RQ($k$-means) direction.

**Codebook enhancements:** A major component of the framework is the codebook, for which we need to adjust the size and number of codebooks. Hence, a natural question is whether we can further improve the model's performance by enhancing codebooks' size, or increasing the number of codes for each input latent vector. This can potentially introduce sparsities in the efficiency of the quantization process and can degrade model performance. We have observed performance degradation in audio and image context if we increase the number of codes, or use *soft codes*, i.e. assigning each input embedding to the top-$k$ closest code vectors based on distance metrics. These results suggest that while large codebooks and soft codes can provide a powerful tokenization process, they may not provide performance benefits in our framework. This outcome may result from the dilution of meaningful features across too many code vectors, or the limitation in the model's capability to learn a large space of tokens, given an encoding embedding of dimension 768 from our base ViT architecture. Additionally, existing codebook methods assume hard-assigned code, and uniform weighting method is shown ineffective. A weighted combination of a sparse subset of codes could be a solution worth exploring in future works (Aharon et al., 2006). Therefore, determining an optimal codebook size is crucial.

**Is joint training of the encoder and the tokenizer possible?** A disadvantage of the BRIDLE pretraining framework is the long training process, where we interchangeably train the encoder $E$ and tokenizer $T$. To accelerate training, we consider a joint training strategy within BRIDLE, where the encoder $E$ and tokenizer $T$ are updated simultaneously within each iteration (Grill et al., 2020). This process allows the model components to co-adapt, fostering faster convergence. The loss functions of the two training phases can be combined as follows:

$$\mathcal{L} = \mathcal{L}_{\text{encoder}} + \alpha \mathcal{L}_{\text{tokenizer}}, \tag{8}$$

where $\alpha$ is a loss ratio between the encoder training and tokenizer training. In BEATs' training framework, the main encoder and the tokenizer are trained in a leaved way: update one while the other is frozen (Chen et al., 2023). Additionally, in every bidirectional training iteration, the tokenizer including its codebooks are reset, with the intuition of improving codebook usage and audio feature capturing. In our empirical results, training all architectures simultaneously provides almost equivalent results. Furthermore, to stabilize codebook training, it is beneficial to update the tokenizer for every few steps of training the main encoder. We have observed that $\alpha = 0.5$ and updating the tokenizer every 5 steps works well for VQ. Nevertheless, we find difficulties in tuning the hyperparameters for RQ models to perform well on downstream tasks, despite the codebooks converge nicely. Hence, we leave this idea for future work.

**Best practices for codebook training:** Our empirical studies identified several techniques that enhance codebook training within the BRIDLE framework:

*1. EMA updates*: The EMA update for codebooks demonstrates higher performance than standard backpropagation.

*2. k-means initialization*: Initializing codebooks using k-means clustering significantly outperforms uniform initialization. This method ensures that code vectors are better aligned with the data distribution from the outset.

*3. Input embedding normalization*: Normalizing input embeddings at the beginning of every codebook layer helps stabilize training and improves quantization quality by maintaining consistent scaling across embeddings.

**Codebook evaluations:** To validate the effectiveness of codebook's representations, we evaluate the codebooks on *Code Usage Rate* (CUR), and *Effective Code Usage* (ECU) (See Appendix G for formulations). These metrics help to systematically assess the diversity and balance of code usage across different configurations. We observe that RQ has superior CUR of ~100%, which intuitively can be easier to achieve with smaller size codebooks compared to VQ. It can be additionally observed that $k$-means initialization can generally yields ~100% CUR without training the codebooks. Furthermore, in the ECU metric, RQ shows better performance than VQ, demonstrating better code utilization.

# 6 Conclusion and Future Work

In this paper, we introduced *BRIDLE*, a self-supervised pretraining framework that incorporates residual quantization into the bidirectional encoder paradigm. Using multiple codebooks hierarchically, our approach enables finer discretization of the latent space, enhancing representation quality across audio, image, and video modalities.

Our experiments demonstrated that BRIDLE consistently outperforms traditional vector quantization methods in both fine-tuning and linear probing evaluations. Specifically, we achieved state-of-the-art results on audio classification benchmarks on AudioSet, and showed competitive results on the ImageNet-1K and Kinetics-400 datasets for image and video classification tasks, respectively. The incorporation of residual quantization not only improved performance but also enhanced the generalization ability of the encoder representations.

We conducted a comprehensive analysis of codebook training within the BRIDLE framework, identifying effective strategies such as $k$-means initialization, input embedding normalization, and the use of exponential moving average updates for codebook vectors. These techniques contributed to better codebook utilization and stability during training. By addressing these directions, we aim to further enhance the capabilities of self-supervised learning frameworks and contribute to the development of more robust and efficient models across diverse modalities. While BRIDLE shows promising results, there are several avenues for future exploration, such as joint training framework for efficiency, enhanced codebook learning, or apply the framework in cross-modalities settings.

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

# BRIDLE: Generalized Self-supervised Learning with Quantization

**Appendix**

The Appendix is organized as follows.

- In Appendix A, we provide the details of the hyperparameters.

- In Appendix B, we show details of the datasets we use for each mode.

- In Appendix C, we provide the ablation and sensitivity studies to support hyperparameter choices.

- In Appendix D, we present the details of the EMA update for the codebooks.

- In Appendix E, we discuss the model configurations.

- In Appendix F, we discuss the framework performance in comparison with existing self-supervised learning frameworks on ImageNet.

- In Appendix G, we provide the definitions of some metrics that are used in codebook and tokenizer evaluation.

## A    Hyperparameter Details

Table 5 presents the details of the hyperparameters used in the training of our BRIDLE model.

## B    Datasets

In this section, we show details of the datasets we conduct the experiments for each mode.

**Audio**: We use the *AudioSet* dataset (Gemmeke et al., 2017), which contains over 2 million 10-second sound clips drawn from YouTube videos and labeled across 527 classes. For AudioSet, we collect the data from a 2023 source, for which due to changes in YouTube content availability, approximately 10–15% of the data is no longer accessible, affecting both the training and evaluation sets.

**On the fairness of comparisons under reduced AudioSet-2M.**   The 15% loss of samples reflects YouTube-side video unavailability — uploads that were deleted, made private, restricted, or geo-blocked between AudioSet's original release (2017) and our scrape (2023) — and is therefore a *temporal-availability artifact* rather than a content-conditional sampling. The mechanism by which a video becomes unavailable (uploader deletion, account closure, copyright takedown, geo-blocking, age-restriction) is governed by uploader actions and platform policy rather than by the audio content itself, so there is no direct mechanism by which the loss would correlate with audio class. We provide two complementary lines of evidence supporting that the loss is not content-driven, and clarify why the comparisons reported in Table 2 remain internally controlled.

*(i) Cross-segment uniformity.* Table 6 reports loss rates of 15.7%, 14.8%, and 15.9% on the Balanced train, Unbalanced train, and Evaluation segments, respectively — three independently sampled subsets of AudioSet that have substantially different class mixes (the Balanced and Evaluation segments each contain at least one example per class by construction, while the Unbalanced segment is dominated by frequent classes). The three loss rates agree to within 1.1 percentage points. If the loss mechanism were strongly content-conditional, segments with different class mixes would show larger divergence in their overall loss rate. The tight agreement across segments is consistent with a content-agnostic loss process.

*(ii) Within-table controlled comparisons.* All rows of Table 2, including the BEATs reproduction, are trained and evaluated on the *same* reduced AudioSet-2M subset. The within-Table-1 comparison between (VQ) BEATs and (RQ) BRIDLE is therefore internally controlled — both rows see the identical data — so any data-loss-induced shift in absolute mAP cancels out in the contrast that supports our main claim. The $\sim 0.5\%$ mAP drop discussed in Section 4.3 is the BEATs-paper-vs-our-BEATs-reproduction gap on full vs. reduced

Table 5: Hyperparameters for the BRIDLE Model across different modalities

| Hyperparameter | Audio AS-2M | AS-20K | ESC-50 | Image ImageNet-1K | Video Kinetics-400 |
|---|---|---|---|---|---|
| Pretrain Batch Size ($B$) | | 16 | | 16 | 32 |
| FT & LP Batch Size ($B$) | | 8 | | 32 | 32 |
| Encoder Pretrain LR | | $5 \times 10^{-4}$ | | $5 \times 10^{-4}$ | $1.2 \times 10^{-3}$ |
| Tokenizer Pretrain LR | | $2 \times 10^{-4}$ | | $2 \times 10^{-4}$ | $2.5 \times 10^{-4}$ |
| FT LR | $5 \times 10^{-4}$ | $4 \times 10^{-4}$ | $2 \times 10^{-3}$ | $5 \times 10^{-4}$ | $7.5 \times 10^{-4}$ |
| LP LR | $4 \times 10^{-3}$ | $4 \times 10^{-2}$ | — | $2 \times 10^{-3}$ | — |
| Encoder Pretrain Epochs | | 130 | | 400 | 800 |
| Tokenizer Pretrain Epochs | | 30 | | 100 | 200 |
| FT Epochs | 120 | 150 | 300 | 200 | 75 |
| LP Epochs | 250 | 400 | — | 300 | — |
| Pretrain Mask Ratio | | 0.8 | | 0.8 | 0.9 |
| FT Masking | | 2D, ratio 0.2 | | 1D, ratio 0.2 | 0.0 |
| LP Masking | 0.0 | 0.0 | — | 0.0 | — |
| Optimizer | | | AdamW | | |
| Weight Decay | | | $1 \times 10^{-4}$ | | |
| FT Loss | | BCE | | BCE | CE |
| LP Loss | BCE | BCE | — | BCE | — |
| Normalization Mean | $-4.4446096$ | $-4.4446096$ | $-6.6268077$ | ImageNet default mean | |
| Normalization Std | 3.3216383 | 3.3216383 | 5.358466 | ImageNet default std | |
| Codebook Num ($M$), VQ | | | 1 | | |
| Codebook Size ($K_m$), VQ | | | 1024 | | |
| Codebook Num ($M$), RQ | | | 4 | | |
| Codebook Size ($K_m$), RQ | | | 256 | | |
| Codebook Dim ($D$) | | | 256 | | |
| EMA Decay Rate ($\gamma$) | | | 0.99 | | |
| Pretrain nnodes | | | 8 | | |
| FT & LP nnodes | 4 | 4 | 1 | 4 | 8 |
| GPU per Node | | | 8 | | |
| Code Reset Threshold ($T_u$) | | | 1 | | |

Table 6: Data availability in AudioSet segments

| Data Segment | Original | Obtained | Percentage |
|---|---|---|---|
| Balanced train | 22,160 | 18,685 | 84.3% |
| Unbalanced train | 2,041,789 | 1,738,788 | 85.2% |
| Evaluation | 20,371 | 17,142 | 84.1% |

data; it is *not* the BEATs-vs-BRIDLE gap reported in Table 2. We acknowledge that absolute numbers in Table 2 should not be compared directly to BEATs as published on the original AudioSet-2M; the controlled BEATs reproduction row provides the appropriate apples-to-apples reference for our setting. We additionally note that since the controlled BEATs reproduction itself trails the original BEATs by only $\sim 0.5\%$ mAP — well within the variance one would expect from any content-correlated bias — the practical impact of the missingness on relative benchmark conclusions is limited.

This data loss may introduce variations in absolute performance metrics. Additionally, we use the *ESC-50* dataset (Piczak, 2015), a collection of 2,000 5-second environmental audio recordings across 50 classes, to evaluate the model's generalization ability in environmental sound classification.

**Image**: For image data, we utilize the *ImageNet-1K* dataset (Deng et al., 2009), which consists of approximately 1.28 million training images and 50,000 validation images in 1,000 classes. ImageNet-1K is a

standard benchmark for image classification tasks and enables us to evaluate the model's performance in visual recognition.

**Video**: For video data, we employ the *Kinetics-400* dataset (Kay et al., 2017), which contains around 240,000 video clips annotated with 400 human action classes. Each clip is approximately 10 seconds long and is sourced from YouTube videos. This dataset allows us to assess the model's capability in capturing spatiotemporal information for action recognition tasks. For our setup, the video frames have dimensions 512 (W) × 288 (H).

## C Ablation and Sensitivity Studies

In the following tables, we provide sensitivity studies with variations of training strategies we tested to arrive at our optimal choices stated in Appendix A. For each table, we use our best results for audio or image experiments at a specific iteration as the benchmark, then state the metric changes as the training strategies vary. "—" represents the same hyperparameter or strategy as the benchmark.

Table 7: BRIDLE sensitivity experiments on AudioSet-2M at iteration 1

| Codebooks params | Codebook init | normalize codebooks | soft code | imagenet pretrained | FT mAP % chg | LP mAP % chg |
|---|---|---|---|---|---|---|
| 4 sets, 256 codes | kmeans | yes | no | no | 0 (47.17) | 0 (29.58) |
| — | — | — | 4 | — | −0.6 | −2.37 |
| 3 sets, 256 codes | — | no | — | — | −0.4 | −0.86 |
| — | — | no | — | — | −0.11 | −1.01 |
| — | uniform | — | — | — | +0.02 | −0.43 |
| — | uniform | — | no | — | −0.03 | −3.66 |
| 1 set, 1024 codes | uniform | — | no | yes | −0.33 | −9.24 |

Table 8: BRIDLE sensitivity experiments on AudioSet-2M at iteration 2

| Codebooks params | Codebook init | normalize codebooks | codebook training strategy | FT mAP % chg | LP mAP % chg |
|---|---|---|---|---|---|
| 4 sets, 256 codes | kmeans | yes | EMA | 0 (47.99) | 0 (29.56) |
| — | — | no | — | −0.54 | −2.48 |
| — | uniform | — | — | −0.43 | −0.85 |
| — | uniform | no | — | −0.41 | −1.93 |
| — | uniform | no | back prop | −0.65 | −3.05 |
| 1 set, 1024 codes | uniform | no | — | −0.35 | −2.52 |

Table 9: BRIDLE sensitivity experiments on ImageNet-1K at iteration 1

| Codebooks params | Codebook init | normalize codebooks | loss function | FT Acc Top-1 % chg | FT Acc Top-5 % chg |
|---|---|---|---|---|---|
| 4 sets, 256 codes | kmeans | yes | BCE | 0 (80.00) | 0 (94.30) |
| — | — | — | CE | −0.97 | +0.17 |
| 4 sets, 512 codes | — | no | — | −0.21 | −0.09 |

## D The EMA Update for the Codebooks

To maintain stable updates of the codebook vectors, we employ the EMA strategy (van den Oord et al., 2017). For each code vector $\mathbf{c}_i$ in a codebook $\mathcal{C}_m$, the update rule at every time step $t$ is defined as follows. We state the result for a generic sequence of latent vectors $\{\mathbf{z}_{j,t}\}_{j=1}^{S}$ feeding the codebook; in the BRIDLE framework, this sequence is the tokenizer-encoder output $\{\tilde{\mathbf{z}}_{j,t}\}_{j=1}^{S}$ from Section 3, but the convergence analysis below

Table 10: BRIDLE sensitivity experiments on ImageNet-1K at iteration 2

| Codebooks params | Codebook init | normalize codebooks | loss function | FT Acc Top-1 % chg | FT Acc Top-5 % chg |
|---|---|---|---|---|---|
| 4 sets, 256 codes | kmeans | yes | BCE | 0 (80.00) | 0 (94.30) |
| — | — | — | CE | $-1.48$ | $-0.07$ |

depends only on properties of the latent sequence (boundedness and almost-sure convergence) and so applies verbatim to either symbol; we use the unsubscripted $\mathbf{z}$ in this appendix purely for notational simplicity. [1]

$$n_{i,t} := \sum_{j=1}^{S} \delta_{q_{j,t},i}, \qquad \ell_{i,t} := \sum_{j=1}^{S} \delta_{q_{j,t},i} \cdot \mathbf{z}_{j,t}, \tag{9}$$

$$N_{i,t+1} \leftarrow \gamma N_{i,t} + (1-\gamma)n_{i,t}, \tag{10}$$

$$\hat{N}_{i,t+1} \leftarrow (N_{i,t+1} + \epsilon)\frac{\sum_{i=1}^{K_m} N_{i,t+1}}{\sum_{i=1}^{K_m} N_{i,t+1} + K_m \cdot \epsilon}, \tag{11}$$

$$\mathbf{m}_{i,t+1} \leftarrow \gamma \mathbf{m}_{i,t} + (1-\gamma)\ell_{i,t}, \tag{12}$$

$$\mathbf{c}_{i,t+1} \leftarrow \frac{\mathbf{m}_{i,t+1}}{\hat{N}_{i,t+1}}, \tag{13}$$

where:

- $\gamma \in (0,1)$ is the EMA decay rate.

- $S$ is the total number of samples (sample size) at every time step.

- $\delta_{q_{j,t},i}$ is the Kronecker delta function, which equals to 1 if $q_{j,t} = i$ and 0 otherwise, where $q_{j,t}$ represents the code that is mapped to the latent vector $\mathbf{z}_{j,t}$.

- $\mathbf{z}_{j,t}$ is the latent vector of sample $j$ at time $t$.

- $N_{i,t}$ is the accumulated usage count (cluster size) of the code vector $\mathbf{c}_i$, where $N_{i,0} = 0$.

- $\mathbf{m}_{i,t}$ is the running embedding update for code vector $\mathbf{c}_{i,t}$, where $\mathbf{m}_{i,0} = \mathbf{c}_{i,0} = \mathbf{c}_0$ for some vector $\mathbf{c}_0$ of dimension $D$.

The EMA update can also be used to stabilize the learning of the codebook vectors in the Residual Quantization (RQ) process that can work directly with mini-batches. Each set of codebook $\mathcal{C}_m$ for $m \in \{1, 2, \ldots, M\}$ can be updated by the above update rule.

Mathematically, the EMA update (9)-(13) can be re-written as

$$N_{i,t+1} = \gamma N_{i,t} + (1-\gamma)n_{i,t} + \epsilon, \tag{14}$$

$$\mathbf{m}_{i,t+1} = \gamma \mathbf{m}_{i,t} + (1-\gamma)\ell_{i,t}, \tag{15}$$

$$\mathbf{c}_{i,t+1} = \frac{\mathbf{m}_{i,t+1}}{\gamma N_{i,t} + (1-\gamma)n_{i,t} + \epsilon} \cdot \frac{\sum_{i=1}^{K_m}(\gamma N_{i,t} + (1-\gamma)n_{i,t}) + K_m \cdot \epsilon}{\sum_{i=1}^{K_m}(\gamma N_{i,t} + (1-\gamma)n_{i,t})}, \tag{16}$$

for any $t = 0, 1, 2, \ldots$ with $N_{i,0} = 0$, $\mathbf{m}_{i,0} = \mathbf{c}_{i,0} = \mathbf{c}_0$, where $n_{i,t}, \ell_{i,t}$ are given in (9).

Next, we provide convergence analysis for the EMA update (9)-(13). Before we proceed, let us show that $\mathbf{m}_{i,t}$ and $N_{i,t}$ are uniformly bounded in $t$, which will be used in our convergence analysis. Indeed, we will first show the following technical lemma for $\mathbf{m}_{i,t}$.

---

[1]Note that $\mathbf{c}_i$ depends on $m$. To simplify the notation, we write $\mathbf{c}_i$ instead of $\mathbf{c}_i^{(m)}$ to hide the dependence on $m$. Similarly, for the update rule for $n_{i,t}, N_{i,t}, \hat{N}_{i,t}, \mathbf{m}_{i,t}, \mathbf{c}_{i,t}$ in (9)-(13), we hide the dependence on $m$ to ease the notation.

**Lemma 1.** *Assume that for every $j = 1, \ldots, S$, $\sup_t \|\mathbf{z}_{j,t}\| < \infty$ almost surely. Then, for any $i = 1, 2, \ldots, K_m$ and $t = 0, 1, 2, \ldots$,*

$$\|\mathbf{m}_{i,t}\| \leq \mathcal{B}_{\mathbf{m}}, \tag{17}$$

*almost surely, where*

$$\mathcal{B}_{\mathbf{m}} := \|\mathbf{c}_0\| + \sum_{j=1}^{S} \sup_t \|\mathbf{z}_{j,t}\|. \tag{18}$$

*Proof.* We use mathematical induction to prove (17). First, when $t = 0$, $\mathbf{m}_{i,0} = \mathbf{c}_0$ so that (17) trivially holds. Next, let us assume that $\|\mathbf{m}_{i,t}\| \leq \mathcal{B}_{\mathbf{m}}$ almost surely for every $i = 1, 2, \ldots, K_m$. Then, from (15), we have

$$\|\mathbf{m}_{i,t+1}\| \leq \gamma \|\mathbf{m}_{i,t}\| + (1 - \gamma)\|\ell_{i,t}\|$$

$$\leq \gamma \mathcal{B}_{\mathbf{m}} + (1 - \gamma) \sum_{j=1}^{S} \sup_t \|\mathbf{z}_{j,t}\| \leq \mathcal{B}_{\mathbf{m}}, \tag{19}$$

almost surely, where we used the definition of $\mathcal{B}_{\mathbf{m}}$ in (18). This completes the proof. □

Next, let us show that $N_{i,t}$ is uniformly bounded.

**Lemma 2.** *For any $i = 1, 2, \ldots, K_m$ and $t = 0, 1, 2, \ldots$,*

$$N_{i,t} \leq \mathcal{B}_N, \tag{20}$$

*almost surely, where*

$$\mathcal{B}_N := S + \frac{\epsilon}{1 - \gamma}. \tag{21}$$

*Proof.* First, we recall from (14) that

$$N_{i,t+1} = \gamma N_{i,t} + (1 - \gamma)n_{i,t} + \epsilon. \tag{22}$$

We use mathematical induction to prove (20). First, when $t = 0$, $N_{i,0} = 0 \leq \mathcal{B}_N$ since $\mathcal{B}_N > 0$, so that (20) trivially holds. Next, let us assume that $N_{i,t} \leq \mathcal{B}_N$ almost surely for every $i = 1, 2, \ldots, K_m$. Then, we have

$$N_{i,t+1} \leq \gamma \mathcal{B}_N + (1 - \gamma)S + \epsilon = \gamma S + \frac{\gamma \epsilon}{1 - \gamma} + (1 - \gamma)S + \epsilon = S + \frac{\epsilon}{1 - \gamma} = \mathcal{B}_N, \tag{23}$$

almost surely, where we used the definition of $\mathcal{B}_N$ in (21) and the definition of $n_{i,t}$ in (9) such that $n_{i,t} \leq S$ for any $i$ and $t$. Hence, we proved (20). □

Now, we are ready to show the convergence of the EMA update (9)-(13). We have the following result.

**Proposition 3.** *Assume that for every $j = 1, 2, \ldots, S$, $q_{j,t}$ and $\mathbf{z}_{j,t}$ converge almost surely as $t \to \infty$. Then, $(N_{i,t}, \mathbf{m}_{i,t}, \mathbf{c}_{i,t})$ converges almost surely to $(N_{i,\infty}, \mathbf{m}_{i,\infty}, \mathbf{c}_{i,\infty})$ as $t \to \infty$ for every $i = 1, 2, \ldots, K_m$, where*

$$\mathbf{m}_{i,\infty} = \ell_{i,\infty},$$

$$N_{i,\infty} = n_{i,\infty} + \frac{\epsilon}{1 - \gamma},$$

$$\mathbf{c}_{i,\infty} = \frac{\ell_{i,\infty}}{n_{i,\infty} + \frac{\gamma\epsilon}{1-\gamma} + \epsilon} \frac{\sum_{i=1}^{K_m}(n_{i,\infty} + \frac{\gamma\epsilon}{1-\gamma} + \epsilon)}{\sum_{i=1}^{K_m}(n_{i,\infty} + \frac{\gamma\epsilon}{1-\gamma})},$$

*with $\ell_{i,\infty} = \sum_{j=1}^{S} \delta_{q_{j,\infty},i} \cdot \mathbf{z}_{j,\infty}$ and $n_{i,\infty} = \sum_{j=1}^{S} \delta_{q_{j,\infty},i}$.*

*Proof.* We prove convergence of $\mathbf{m}_{i,t}$ and $N_{i,t}$ along the entire iterate sequence by a direct contraction (vanishing-perturbation) argument applied to the EMA recurrences. Convergence of $\mathbf{c}_{i,t}$ then follows by the continuous mapping theorem.

**Step 1: convergence of $\mathbf{m}_{i,t}$.** For every $j = 1, \ldots, S$, $\mathbf{z}_{j,t}$ converges almost surely, so it is bounded almost surely; the hypothesis of Lemma 1 is satisfied and $\mathbf{m}_{i,t}$ is uniformly bounded a.s. in $t$. Moreover, since $q_{j,t}$ and $\mathbf{z}_{j,t}$ converge a.s., we have $\ell_{i,t} \to \ell_{i,\infty}$ a.s. as $t \to \infty$, where $\ell_{i,\infty} = \sum_{j=1}^{S} \delta_{q_{j,\infty},i} \cdot \mathbf{z}_{j,\infty}$.

Define

$$a_t := \left\| \mathbf{m}_{i,t} - \ell_{i,\infty} \right\|, \qquad \eta_t := (1 - \gamma) \left\| \ell_{i,t} - \ell_{i,\infty} \right\|, \tag{24}$$

so that $\eta_t \to 0$ a.s. From the recurrence (15) and the identity $\ell_{i,\infty} = \gamma \ell_{i,\infty} + (1 - \gamma) \ell_{i,\infty}$, we obtain

$$\mathbf{m}_{i,t+1} - \ell_{i,\infty} = \gamma \left( \mathbf{m}_{i,t} - \ell_{i,\infty} \right) + (1 - \gamma) \left( \ell_{i,t} - \ell_{i,\infty} \right), \tag{25}$$

which, after taking norms, yields

$$a_{t+1} \le \gamma \, a_t + \eta_t. \tag{26}$$

Iterating (26) from $t = 0$ gives, for any $t \ge 1$,

$$a_t \ \le \ \gamma^t \, a_0 \ + \ \sum_{s=0}^{t-1} \gamma^{t-1-s} \, \eta_s. \tag{27}$$

The first term in (27) satisfies $\gamma^t a_0 \to 0$ since $\gamma \in (0, 1)$ and $a_0$ is finite. We now show the second term tends to zero. Fix any $\delta > 0$. Since $\eta_s \to 0$ a.s., there exists (a.s.) a finite $T_0(\delta)$ such that $\eta_s < \delta$ for all $s \ge T_0$. For $t > T_0$, split

$$\sum_{s=0}^{t-1} \gamma^{t-1-s} \, \eta_s \ = \ \underbrace{\sum_{s=0}^{T_0-1} \gamma^{t-1-s} \, \eta_s}_{(I)} \ + \ \underbrace{\sum_{s=T_0}^{t-1} \gamma^{t-1-s} \, \eta_s}_{(II)}. \tag{28}$$

For term $(I)$, since $\eta_s$ is uniformly bounded by $C := \sup_s \eta_s < \infty$ a.s. (a consequence of Lemma 1 applied to the bounded sequence $\ell_{i,t}$),

$$(I) \ \le \ C \sum_{s=0}^{T_0-1} \gamma^{t-1-s} \ \le \ C \gamma^{t-T_0} \cdot \frac{1 - \gamma^{T_0}}{1 - \gamma} \ \xrightarrow[t \to \infty]{} \ 0,$$

since $\gamma^{t-T_0} \to 0$ as $t \to \infty$. For term $(II)$,

$$(II) \ \le \ \delta \sum_{s=T_0}^{t-1} \gamma^{t-1-s} \ \le \ \frac{\delta}{1 - \gamma}.$$

Combining, $\limsup_{t \to \infty} a_t \le \delta/(1 - \gamma)$ a.s.; since $\delta > 0$ is arbitrary, $a_t \to 0$ a.s. Therefore

$$\mathbf{m}_{i,t} \ \xrightarrow{\text{a.s.}} \ \ell_{i,\infty} \ =: \ \mathbf{m}_{i,\infty} \quad \text{as} \quad t \to \infty, \tag{29}$$

along the *entire* sequence, establishing the claimed limit $\mathbf{m}_{i,\infty} = \ell_{i,\infty}$.

**Step 2: convergence of $N_{i,t}$.** The argument is structurally identical. Since $q_{j,t}$ converges a.s., $n_{i,t} = \sum_{j=1}^{S} \delta_{q_{j,t},i} \to n_{i,\infty} = \sum_{j=1}^{S} \delta_{q_{j,\infty},i}$ a.s. as $t \to \infty$. Define $N_{i,\infty}^* := n_{i,\infty} + \frac{\epsilon}{1 - \gamma}$, which is the unique fixed point of (14) when $n_{i,t} = n_{i,\infty}$. Set

$$b_t := \left| N_{i,t} - N_{i,\infty}^* \right|, \qquad \xi_t := (1 - \gamma) \left| n_{i,t} - n_{i,\infty} \right|,$$

with $\xi_t \to 0$ a.s. (in fact, since $n_{i,t}$ is integer-valued and converges a.s., $\xi_t = 0$ for all $t$ sufficiently large). From (14),

$$N_{i,t+1} - N_{i,\infty}^* = \gamma (N_{i,t} - N_{i,\infty}^*) + (1 - \gamma)(n_{i,t} - n_{i,\infty}),$$

hence $b_{t+1} \le \gamma b_t + \xi_t$. The same unrolling argument as in Step 1 (replace $a_t, \eta_t$ by $b_t, \xi_t$) yields

$$N_{i,t} \ \xrightarrow{\text{a.s.}} \ N_{i,\infty}^* = n_{i,\infty} + \frac{\epsilon}{1 - \gamma} \quad \text{as} \quad t \to \infty, \tag{30}$$

along the entire sequence, for every $i = 1, 2, \ldots, K_m$.

**Step 3: convergence of $\mathbf{c}_{i,t}$.** We recall from (16) that

$$\mathbf{c}_{i,t} = \frac{\mathbf{m}_{i,t}}{\gamma N_{i,t-1} + (1-\gamma)n_{i,t-1} + \epsilon} \cdot \frac{\sum_{i=1}^{K_m}(\gamma N_{i,t-1} + (1-\gamma)n_{i,t-1}) + K_m \cdot \epsilon}{\sum_{i=1}^{K_m}(\gamma N_{i,t-1} + (1-\gamma)n_{i,t-1})}. \tag{31}$$

For every $i$, the denominator $\gamma N_{i,t-1} + (1-\gamma)n_{i,t-1} + \epsilon \geq \epsilon > 0$ is uniformly bounded away from zero, and the second-factor denominator $\sum_{i=1}^{K_m}(\gamma N_{i,t-1} + (1-\gamma)n_{i,t-1}) \geq 0$ is bounded as the right-hand-side denominator stays positive provided that $\sum_{i=1}^{K_m} n_{i,t-1} > 0$ (i.e., at least one assignment occurs at every step), which holds by the definition of $n_{i,t}$ on a non-empty mini-batch. By Steps 1 and 2 together with the continuous mapping theorem, $\mathbf{c}_{i,t} \to \mathbf{c}_{i,\infty}$ a.s. as $t \to \infty$, where

$$\begin{aligned}
\mathbf{c}_{i,\infty} &= \frac{\mathbf{m}_{i,\infty}}{\gamma N_{i,\infty} + (1-\gamma)n_{i,\infty} + \epsilon} \cdot \frac{\sum_{i=1}^{K_m}(\gamma N_{i,\infty} + (1-\gamma)n_{i,\infty}) + K_m \cdot \epsilon}{\sum_{i=1}^{K_m}(\gamma N_{i,\infty} + (1-\gamma)n_{i,\infty})} \\
&= \frac{\ell_{i,\infty}}{n_{i,\infty} + \frac{\gamma\epsilon}{1-\gamma} + \epsilon} \frac{\sum_{i=1}^{K_m}(n_{i,\infty} + \frac{\gamma\epsilon}{1-\gamma} + \epsilon)}{\sum_{i=1}^{K_m}(n_{i,\infty} + \frac{\gamma\epsilon}{1-\gamma})},
\end{aligned} \tag{32}$$

where in the second equality we used $\mathbf{m}_{i,\infty} = \ell_{i,\infty}$ and $N_{i,\infty} = n_{i,\infty} + \epsilon/(1-\gamma)$, so that $\gamma N_{i,\infty} + (1-\gamma)n_{i,\infty} = n_{i,\infty} + \gamma\epsilon/(1-\gamma)$. The proof is complete. $\square$

**Remark 4.** *The replacement of the original Bolzano–Weierstrass / subsequential argument by the contraction-based proof above resolves a subtle issue: a uniformly bounded sequence is only guaranteed to have a convergent subsequence, and substituting that subsequential limit into a recurrence indexed by the full sequence (such as (15)) does not, on its own, identify the limit of the entire sequence. The vanishing-perturbation form of (26), together with $\eta_t \to 0$, eliminates the need for any such substitution and directly yields convergence along the entire iterate sequence. We thank an anonymous reviewer for surfacing this point.*

## E  Model Configurations

Our experiments use the base ViT architecture for audio and image, and the base video ViT architecture for video mode.

**Audio and Image**: For audio and image modalities, we follow the configurations of BEATs (Chen et al., 2023) and AudioMAE (Huang et al., 2022). The encoder architecture is based on a transformer model with appropriate modifications to handle mel-spectrograms for audio and patches for images.

**Video**: For the video modality, we adopt the setup from VideoMAE (Tong et al., 2022). The encoder is extended to process spatiotemporal data by incorporating temporal attention mechanisms to capture motion dynamics effectively.

The decoder is a 16-block Swin transformer for audio and image, and a 4-block transformer for video. The tokenizer estimator is a simple 3-block transformer for audio and image, and a 4-block transformer for video.

Each of $M = 4$ codebooks for RQ has a size of $K_m = 256$ code vectors of dimension $D = 256$, initialized based on the input latent vectors with $k$-means initialization of 10 steps, or with random uniform initialization. Meanwhile, the VQ version of the model has $M = 1$ codebook of $K_m = 1024$ code vectors also of dimension $D = 256$. Hence, the VQ and RQ versions of the training framework contains the same number of codes, where each latent vector is mapped to $M = 1$ code for VQ, and $M = 4$ codes for RQ. The base ViT encoder and tokenizer encoder produce 768-dimensional features, which are projected to the $D = 256$ codebook dimension before quantization.

## F  Benchmark Comparisons on ImageNet

Although BRIDLE ranks below the masked-image-modeling family by 2–3 pp in Top-1 accuracy, its design offers several advantages that can make this trade-off attractive:

Table 11: Fine–tuned ImageNet-1K accuracy for self-supervised models.

| Method | Top-1 Acc (%) |
|---|---|
| ViT384-B-JFT300M (Dosovitskiy et al., 2021) | 79.9 |
| BRIDLE (ours) | 81.1 |
| DINO (Caron et al., 2021) | 82.8 |
| BEiT (Bao et al., 2022) | 83.2 |
| MoCo v3 (Chen et al., 2021) | 83.2 |
| MAE (He et al., 2022) | 83.6 |

1. Training efficiency: BRIDLE reaches its reported score after only two bidirectional iterations, each markedly shorter than the 800–1600 epoch schedules typical for MAE or BEiT.

2. Discrete token output: Thanks to residual quantisation, BRIDLE supplies high-entropy visual tokens directly from the encoder, eliminating the need for a separate tokenizer when the features will feed downstream generative or retrieval models.

3. Modality-agnostic recipe: The same encoder–tokenizer framework has been shown to transfer to audio and video without architectural changes, yielding consistent gains over VQ baselines across modalities.

4. Label-efficient features: In linear probing, BRIDLE outperforms its VQ counterpart, indicating that its representations capture semantically useful structure even when only a small fraction of ImageNet labels is available.

Overall, BRIDLE offers compute-conscious pre-training that natively outputs discrete tokens, making it an appealing foundation when cross-modal alignment, low-resource fine-tuning, or downstream generative tasks are the priority.

## G  Codebook and Tokenizer Evaluation

To evaluate the effectiveness of the codebooks in the BRIDLE model's tokenizer, we use several metrics that assess both the diversity of code usage and the balance of their usage. These metrics help ensure that the tokenizer effectively utilizes the entire codebook, leading to robust audio representations.

**Code Usage Rate (CUR):** Measures the proportion of codes within every codebook $\mathcal{C}_m$ that are used at least once during tokenization:

$$\text{CUR} = \frac{\text{Number of unique codes used}}{K_m}. \tag{33}$$

A high CUR value indicates diverse usage of the codebook.

**Usage Entropy (UE):** Quantifies the uniformity of code usage using Shannon entropy (Shannon, 1948):

$$\text{UE} = -\sum_{i=1}^{K_m} p_i \log(p_i), \tag{34}$$

where $p_i = \frac{\hat{N}_{i,t}}{\sum_{j=1}^{K_m} \hat{N}_{j,t}}$ is the probability of selecting the $i$-th code, with $\hat{N}_{i,t}$ being the accumulated (normalized) usage count of the $i$-th code across the entire dataset (the $\epsilon$-regularized count $\hat{N}$ of Appendix D). A higher UE suggests more balanced code usage.

**Effective Code Usage (ECU):** Combines CUR (33) and UE (34) to evaluate both the diversity and balance of code usage:

$$\text{ECU} = \text{CUR} \times \frac{\text{UE}}{\log(K_m)}. \tag{35}$$

In Table 12, we show ∼100% CUR for pre and post-training RQ. Additionally, the table shows superior ECU of RQ compared to VQ. A well-spread distributions of code usage is also demonstrated in Figure 3 for RQ codebooks.

Table 12: CUR and ECU for different methods before and after training

| Iteration | Method | Codebook # | CUR ↑ | ECU ↑ |
|---|---|---|---|---|
| BRIDLE$_{iter1}$ (random init codebook) | VQ | - | 1.00 | 0.0146 |
| | RQ (uniform) | 1 | 1.00 | 0.0235 |
| | | 2 | 0.99 | 0.0284 |
| | | 3 | 1.00 | 0.0489 |
| | | 4 | 0.86 | 0.0367 |
| | RQ ($k$-means) | 1 | 0.99 | 0.0234 |
| | | 2 | 1.00 | 0.0326 |
| | | 3 | 1.00 | 0.0320 |
| | | 4 | 1.00 | 0.0310 |
| BRIDLE$_{iter2}$ (post training codebook) | VQ | - | 0.21 | 0.0041 |
| | RQ (uniform) | 1 | 0.99 | 0.0275 |
| | | 2 | 1.00 | 0.0231 |
| | | 3 | 1.00 | 0.0228 |
| | | 4 | 1.00 | 0.0229 |
| | RQ ($k$-means) | 1 | 0.99 | 0.0283 |
| | | 2 | 1.00 | 0.0229 |
| | | 3 | 1.00 | 0.0228 |
| | | 4 | 1.00 | 0.0228 |

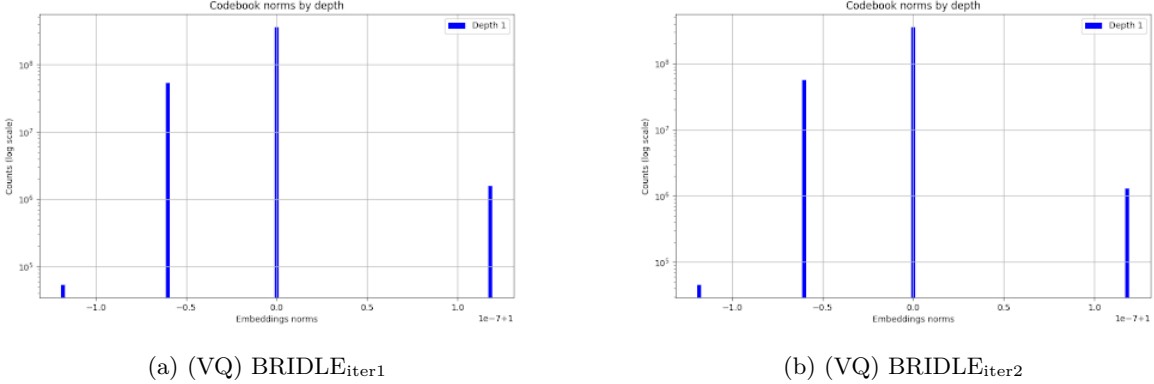

(a) (VQ) BRIDLE$_{iter1}$                (b) (VQ) BRIDLE$_{iter2}$

Figure 2: VQ codebook embeddings norm distributions before and after training on AudioSet

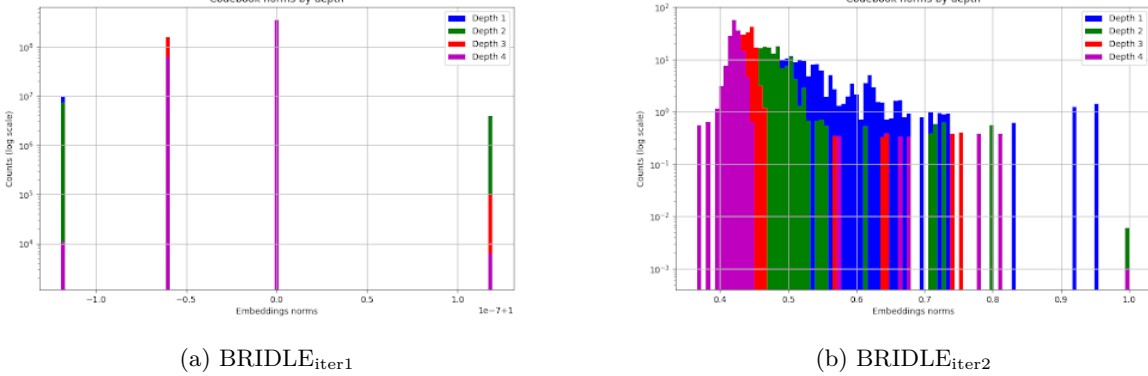

(a) BRIDLE$_{iter1}$                                        (b) BRIDLE$_{iter2}$

Figure 3: RQ codebook embeddings norm distributions before and after training on AudioSet

