# OpenReview forum: "BRIDLE: Generalized Self-supervised Learning with Quantization"
_TMLR — Under review for TMLR_

### Review · Reviewer_GdKj · 2026-04-15

**Summary Of Contributions:**

To my understanding, the main contributions of this paper are threefold:

1. Introduces a new training framework called BRIDLE using residual quantization technique in the tokenizer based on BEATs framework (Chen et al., 2023).

2. Provides  a convergence analysis of the EMA update for all potential codebooks.

3. Provides comprehensive evaluations of this BRIDLE framework in audio, image and video classification, respectively.

**Audience:**

Yes

**Audience Explanation:**

Some techniques like residual quantization and the bidirectional training paradigm is very hot in the field and it is exciting to share some insights in both theoretical and empirical aspects.

**Broader Impact Concerns:**

No.

**Claims And Evidence:**

Yes

**Claims Explanation:**

However, I made some comments below before giving a positive answer to this question.

1. It seems that there is no noval technique/theory introduced in this paper, but it is interesting to show more details and more motivations/theoretical insights of RQ + BEATs to help TMLR readers to understand the fundamental framework BEATs. Also, there is a disconnect between the notations presented in Section 3 and Fig.1, specifically $z_t$, which is a latent representation produced by Main Encoder $E(;\theta_E)$ according to the text. However, in Equation (5), the authors define the codebook loss $L_{cb}$ using $z_t$, incorporating a commitment loss term:$$L_{cb} = \frac{1}{T} \sum_{t=1}^T \left( \|\text{sg}[z_t] - q_t\|^2 + \beta \|\mathbf{z}_t - \text{sg}[q_t]\|^2 \right),$$ where $z_t$ is mathematically nonsensical in the context of the proposed architecture. I think $z_t$ here is the output of the tokenizer Encoder and the authors must rigorously rectify the functional dependencies and notations throughout Section 3.

2. Regarding the theoretical analysis in Appendix D, I have some mathematical logical doubts about the convergence of the EMA updates $m_{i,t}$ and $N_{i,t}$, which can be clearly seen in the following part. This is also the main point where I am confused.

3. Regarding the experiment results in Section 4.3 and Table 1,the authors explicitly attribute the performance drop of the baseline to "~15% missing data" from the AudioSet-2M dataset. The distribution characteristics of this missing 15% data should be clarified that comparing models on this specific dataset remains a fair and unbiased evaluation results of the proposed architectural improvements.

I would appreciate if the authors could clarify these problems so that I can answer the question positively.

**Requested Changes:**

As I mentioned above, I would request the authors for making a deeper pass on the paper to polish notations and make them consistent through all pages; In Section 3.2, could the author provide us with more explanations on the selection of the loss function? For instance, regarding Eq. (5), could the authors provide more intuition on the necessity of adding the commitment loss in this specific context? And why is the final tokenizer loss presented in the form of Eq. (7)? Providing a brief explanation here would be more beneficial for readers to understand the motivation.


For the theoretical part, the convergence proofs for the EMA update provided in Appendix D contain some algebraic errors and logical jumps:
1. In Eq. (23), the inductive step yields $N_{i,t+1} \le \gamma \mathcal{B}_N + (1-\gamma)S + \epsilon$. Substituting the authors' defined bound $\mathcal{B}_N := S$ gives $S + \epsilon$. The authors incorrectly claim this equals $\mathcal{B}_N$ (i.e., $S + \epsilon = S$). But this error can be corrected in my opinion;
2. The proof of Proposition 3 contains a severe flaw, specifically when transitioning from the existence of a subsequence to applying the recurrence relation in Equation (15). The authors invoke the Weierstrass theorem to correctly state that the bounded sequence $m_{i,t}$ has a convergent subsequence, let us say $m_{i,t_k} \to m_{i,\infty}$ (I use this notation for simplicity). However, to derive Eq. (24), they incorrectly substitute this limit $m_{i,\infty}$ directly into the step-by-step recurrence relation (Eq. 15). And it should be noted that $t_{k+1} \neq t_k + 1$ may not always hold in this case. To take the limit of both sides of Eq. (15), one must evaluate the limit of the entire sequence $\mathbf{m}_{i,t}$, not just a subsequence. At this stage of the proof, it is entirely unknown whether the entire sequence converges; it may have multiple accumulation points. Consequently, the convergence proof is incomplete;


And for the experimental part, presented in Tables 2 and 3, the comparisons are shown exclusively between different variants of the proposed BRIDLE model. Could the authors explain why standard external baselines were not included in these tables?

Overall, this paper has some contributions, but is not yet ready for publication in my opinion. If the authors resolve my doubts, I will change my opinion.

---

> ### Author Response · Authors · 2026-07-04
>
> We thank the reviewer for the careful, technically detailed reading, and especially for identifying two genuine errors in the EMA analysis together with the notation conflict in Eq. (5). We address each point below and indicate where the corresponding change appears in the revised manuscript.
>
> **1. Notation conflict in Eq. (5).** The reviewer is correct that the symbol $\mathbf{z}_t$ was overloaded. We have introduced a distinct symbol $\tilde{\mathbf{z}}_t = E_T(\mathbf{x})$ for the tokenizer-encoder output and reserved $\mathbf{z}_t$ exclusively for the main encoder. With this distinction, the commitment term in the codebook loss acts on the tokenizer encoder rather than implying gradients into the main encoder — the architecturally correct behaviour, since the main encoder is frozen as the teacher during tokenizer training. The corrected notation is propagated through Eqs. (4)–(5), the codebook-training subsection, Figure 1, and the EMA appendix (Appendix D).
>
> **2. Motivation for the loss functions (Sec. 3.2).** We have added two short explanatory paragraphs. The first clarifies the complementary roles of the two codebook-loss terms under the asymmetric stop-gradient pattern: the first term updates the codebooks so they track the tokenizer-encoder feature distribution, while the commitment term keeps the tokenizer encoder close to the codebook entry it is snapped to, preventing it from drifting relative to the slowly-updated codebooks. The second explains why the tokenizer loss combines the codebook and cosine terms — the former controls quantization fidelity within the tokenizer's own representation space, the latter keeps the chosen tokens aligned with the masked-prediction objective the main encoder is trained against.
>
> **3. The $N$-bound (Lemma 2).** The reviewer is right that the inductive step did not close with the stated bound. We corrected the bound to $\mathcal{B}_N := S + \epsilon/(1-\gamma)$, for which the induction now closes exactly. The correction is local, and no subsequent argument relied on the specific value.
>
> **4. Proposition 3.** We agree that substituting a subsequential limit into a full-sequence recurrence does not identify the limit of the entire sequence. We have replaced that step with a direct vanishing-perturbation contraction argument: writing $a_{t+1} \le \gamma a_t + \eta_t$ with $\eta_t \to 0$ and unrolling yields convergence of the running accumulators $m_{i,t}$, the cluster sizes $N_{i,t}$, and — by the continuous mapping theorem — the codebook vectors $c_{i,t}$, each along the entire iterate sequence. The stated limits are unchanged, and we added a remark explaining exactly why the original subsequence step was insufficient. Since the result now holds along the full iterate sequence, we also state it explicitly as a contribution in the introduction.
>
> **5. Fairness of the reduced AudioSet-2M (Appendix B).** We have added an analysis of the ~15% missingness. The loss reflects YouTube-side video unavailability (deletions, privating, geo-blocking) rather than content-conditional sampling, so there is no direct mechanism tying it to audio class. As supporting evidence, the loss rates across three segments with very different class mixes — 15.7%, 14.8% and 15.9% (Table 6) — agree to within 1.1 percentage points, consistent with a content-agnostic process. Most importantly, every audio row, including the BEATs reproduction, is trained and evaluated on the identical reduced subset, so the VQ-vs-RQ comparison is internally controlled; the ~0.5% mAP difference relative to the published BEATs is the gap to our reproduction, not a BEATs-vs-BRIDLE gap.
>
> **6. External baselines in the results tables.** We have promoted external self-supervised baselines from the appendix into the main image and video tables (Tables 3–4): ViT-JFT300M, DINO, BEiT, MoCo v3 and MAE for ImageNet, and VideoMAE for Kinetics-400. We are transparent that on Kinetics-400 BRIDLE trails the video-specialized VideoMAE; this is expected, since BRIDLE applies a single unified audio/image/video recipe with no video-specific tuning, and the claim we make is the controlled within-modality VQ→RQ improvement, which holds across all three modalities.
>
> **7. Notation pass.** We completed a consistency pass on the $\mathbf{z}$ / $\tilde{\mathbf{z}}$ distinction throughout the body and appendices, and verified the lemma and proposition cross-references after the proof rewrite.
>
> We are grateful for such a careful and precise review; the corrections to Lemma 2 and Proposition 3 and the notation cleanup have improved both the rigour and the readability of the paper, and we hope the revision fully addresses the reviewer's concerns.

---

### Review · Reviewer_6oFc · 2026-05-04

**Summary Of Contributions:**

This work presents a pre-training framework, BRIDLE, to integrate, or better operationalize, existing work on residual-quantization (RQ) in the setting of SSL. This shows as an extension of the existing BEATS framework.

The BEATS framework, on which this work is based, uses a VQ with a single codebook containing 1024 embeddings of 256 dimensions. This work replaces this VQ codebook with a single RQ codebook, which, however, is hierarchical and contains 4 intermediate codebooks, in this case, each with 256 (dimensionality) entries.

Both frameworks train the encoder through masked (discrete labels) prediction, where BRIDLE specifically alters how target tokens are generated; namely, by summing the codes from all stages (intermediary codebooks) across all RQ codebooks.

Empirically, the paper shows that RQ improves, or at least matches, the single, albeit larger, codebook of BEATS across audio, image, and video tasks, with the clearest benefits observed during linear probing, showing that the obtained representations are more linearly separable and therefore more suited for downstream tasks.

Finally, the authors examine how codebook settings, including initialization, affect its utilization.

Essentially, the contributions help find better discrete target tokens the encoder can use for fine-tuning by using a hierarchical RQ of the discrete token rather than a single, larger VQ.

Strengths:
- Raising an important question, relevant to the community.
- Addresses/improved the key contribution of the existing BEATS framework, which at the time was central to obtaining SOTA performance in the presented SSL setting.
- Empirical evidence across several settings.

Weaknesses:
- At times, it is hard to distinguish between the contributions of this paper and BEATS.
- Missing justification of issues with single codebooks besides empirical evidence, which previously may have similarly been investigated only superficially.
- Missing focus in the main text, which is on the improvement of useful tokens using RQ instead of VQ, which I see as the most essential contribution, and not that of the framework.

**Additional Comments:**

The paper's narrative would improve as a "better target-tokenizer" than in BEATS, framing, rather than a new framework similar to it. The framework does not provide benefits; it is mostly similar to BEATS.

I recommend a stronger focus on codebook utilization, initialization, and the concentration of codebook values in the main text as a strong contribution, offering insights and for how to obtain useful discrete tokens for downstream tasks.

**Audience:**

Yes

**Audience Explanation:**

A relevant question, previously discussed in the BEATS paper, is posed: how to obtain useful discrete tokens in otherwise continuous audio settings amongst other settings.

The scope is fairly narrow, which I view positively, as the findings isolate a concrete design choice and provide empirical evidence of its impact across multiple modalities.

**Broader Impact Concerns:**

I have no broader impact concerns.

**Claims And Evidence:**

Yes

**Claims Explanation:**

It is reasonably well supported that RQ improves performance over VQ, particularly showing by linear probing. These results are consistent across the modalities presented.

However, some performance gains are marginal, and tables omit standard deviations, especially given the fragility of the codebook initializations in the uniform case, so that some gains may be slightly overstated.

Moreover, as the authors point out that codebook utilization is a problem, which motivates this work, it is not fully established that the performance increases stem from better utilization rather than from the richer space included by the RQ codebook, i.e., the geometry of the representation space.

The paper supports its performance claims reasonably, yet the motivation of the work and a more fundamental understanding of the effect of their approach are limited. This provides little insight into future work and little beyond the, in some cases, performance gains.

**Requested Changes:**

Besides the first and last point, it is believed that all requests are necessarily for acceptance, considering they all pertain fundamentally to the motivation of this work.

- Clarify more clearly the difference between BEATS and BRIDLE, e.g., physically close to each other in the paper. At times, I find it hard to distinguish between the contributions of this work and those of BEATS.
- Reframe the M=4 RQ and M=1 VQ comparison as there are fundamental differences in the methods that impact the effective representation space of the tokens, making it more difficult to compare RQ and VQ just by setting $m \times k$ in BRIDLE equal to embeddings $\times$ dimensionality of BEATS, faithfully.
- Add standard deviation/error bars. Also, the choice of M=4 is not sufficiently justified.
- Provide a clearer explanation of why low codebook utilization is a problem in this setting and whether utilization is actually a cause for better better/worse downstream performance. Recent literature would suggest otherwise, as learning generally biases models towards simpler representations [1, 2, 3]. That is, the geometry of the representation space is more important than its size, and codebooks will likely be underutilized unless they are "large enough".
- Clarify the initialization statement: "uniformly based on the variance of the output features." Why and what does this entail (impact)?
- Minor nitpick is the last row of table 4 being difficult to read in an otherwise nicely formatted table.

[1] Tishby, N., Pereira, F. C., & Bialek, W. (2000). The information bottleneck method. arXiv preprint physics/0004057.

[2] Valle-Perez, G., Camargo, C. Q., & Louis, A. A. (2018). Deep learning generalizes because the parameter-function map is biased towards simple functions. arXiv preprint arXiv:1805.08522.

[3] Wilson, A. G. (2025). Deep learning is not so mysterious or different. arXiv preprint arXiv:2503.02113.

---

> ### Author Response · Authors · 2026-07-04
>
> We thank the reviewer for the framing critique, which was well taken, and for the pointers to the simplicity-bias literature. We address each point below.
>
> **1. Distinguishing BEATs from BRIDLE.** We agree that the earlier draft described BRIDLE component-by-component without making the delta from BEATs concrete. The abstract, introduction, and first contribution now lead with BRIDLE as an improved target tokenizer within the BEATs lineage rather than a new framework. A new Table 1 places the two side by side and marks each component either "same" or the explicit change, and a new Algorithm 1 annotates every step of the training loop as inherited from BEATs or specific to BRIDLE — so the contribution is unambiguous at both the component and the procedural level.
>
> **2. The M=4 versus M=1 matching.** The reviewer is right that equating the atom counts ($M \cdot K_m = 4 \cdot 256 = 1024$) matches the number of atoms but not the effective token space. We now state this explicitly: VQ offers $K_{\mathrm{VQ}} = 1024$ alternatives per token, whereas RQ's additive composition spans up to $K_m^M = 256^4$ structured combinations. We therefore frame the contribution as a richer token geometry rather than more efficient use of a flat codebook, and we spell out the matching condition together with its limits in Section 3 and Section 4.1.
>
> **3. Error bars and the choice of M=4.** On error bars: multi-seed pretraining at our schedule — for AudioSet-2M alone, 130+30 epochs across 8 nodes of 8 GPUs, over two bidirectional iterations — is prohibitively expensive, so we are unable to provide seed-level standard deviations. We have added a "note on variance" that states this plainly and instead grounds the claim in the one setting where the protocol itself produces a spread (ESC-50, reported as a 5-fold cross-validation mean) and in the consistency of the effect: eight independent VQ→RQ comparisons across Tables 2–4 all favour RQ, and the gains are largest under linear probing, where single-run noise is least able to account for them. On M=4: the choice follows from the audio-iter1 ablation (Table 7), where reducing to three codebooks costs 0.4 FT / 0.86 LP mAP, and it matches the four residual stages typical of RQ-VAE-style tokenizers; we did not observe further gains beyond M=4 under the matched-total-codes budget.
>
> **4. Whether utilization is the right thing to optimize.** We agree with the reviewer's point and with the simplicity-bias framing. A new paragraph in Section 5 engages directly with [Tishby et al. 2000; Valle-Perez et al. 2018; Wilson 2025]. Empirically, the VQ codebook collapses to a code-usage rate of about 0.21 after training (Table 12) — roughly 215 of 1024 atoms, exactly the under-utilization simplicity bias predicts — whereas RQ retains near-full usage across its four codebooks. We therefore no longer motivate BRIDLE as "fixing utilization"; instead we argue that RQ provides a better-structured token geometry, of which higher utilization is a downstream symptom rather than the objective.
>
> **5. The initialization statement (Sec. 3.3).** We rewrote this passage to be precise. Because the tokenizer-encoder latents are pre-normalized to approximately unit variance per coordinate on $[-1,1]$, we draw each codebook entry i.i.d. uniform on $[-1,1]^D$, and we state the three-fold intent: bounding entries away from zero, spanning the typical range of the latent, and avoiding a pre-clustered bias. We also make explicit that uniform initialization is the more fragile of the two strategies, which is why $k$-means is the recommended default and uniform is reported only as a sensitivity baseline.
>
> **6. Table formatting.** We fixed the readability of the final row of the hyperparameters table (Table 5).
>
> We are grateful for the framing critique and the simplicity-bias references, which clarified a tension in the original motivation between "fixing utilization" and the linear-probing evidence, and we hope the revision addresses the reviewer's concerns.

---

### Review · Reviewer_u1pk · 2026-06-20

**Summary Of Contributions:**

The paper introduces BRIDLE, a self-supervised learning framework for audio representation learning that extends the BEATs architecture by replacing standard vector quantization (VQ) with residual quantization (RQ). The central idea is that a hierarchy of codebooks can provide a richer and more accurate discretization of audio representations than a single codebook, thereby producing higher-quality training targets for masked prediction. The framework incorporates from BEATs an interleaved training procedure between the tokenizer and encoder, and investigates several techniuqes for improving codebook training.

The experiments show that incorporating residual quantization into the BEATs framework consistently improves performance across audio, image, and video classification tasks. BRIDLE achieves modest but consistent gains over the BEATs baseline on AudioSet and ESC-50. Similar improvements are observed on ImageNet-1K and Kinetics-400, and k-means codebook initialization further enhances results. Overall, the findings support the claim that residual quantization produces higher-quality discrete representations and leads to more robust downstream performance across multiple modalities.

Strengths:
- Simple and well-motivated extension of BEATs using residual quantization.
- Consistent improvements across audio, image, and video benchmarks.
- Useful empirical analysis of codebook initialization and utilization.

Weaknesses:
- Novelty is relatively incremental.
- Gains on audio benchmarks are modest.

**Additional Comments:**

N/A

**Audience:**

Yes

**Audience Explanation:**

- It improves a widely used SSL framework (BEATs-style masked prediction), which is directly relevant to representation learning research.
- It shows consistent gains from a simple but general change (residual quantization) across multiple modalities (audio, image, video).
- It addresses practical issues in vector quantization like codebook collapse.
- It provides empirical evidence about how discrete tokenization choices affect learned representations.

**Claims And Evidence:**

Yes

**Claims Explanation:**

The empirical evidence is mostly consistent and convincing: across audio, image, and video benchmarks, residual quantization (especially with k-means initialization) repeatedly improves both fine-tuning and linear probing results compared to BEATs/VQ variants. The fact that gains also appear in linear probing suggests the improvements are not just from overfitting during fine-tuning, but from stronger learned representations.

However, the empirical evidence does not appear fully clear. The improvements on core audio benchmarks are relatively small, and the paper does not always clearly isolate which components are responsible for the gains (e.g., residual quantization vs. k-means initialization).  Additionally, comparisons outside the BEATs/VQ family (especially for image/video) are less emphasized, so the broader “state-of-the-art across modalities” claim is less strongly supported.

Overall: the claims support "RQ improves BEATs-style SSL", but the evidence is less convincing for attributing gains to specific design choices or for strong cross-domain SOTA claims.

**Requested Changes:**

- Add clearer ablations isolating the effect of residual quantization from k-means initialization, training schedule, and other design choices to better attribute performance gains.
- Provide a more explicit algorithmic description and comparison to BEATs-style training to make clear whether the interleaving strategy is genuinely new or mainly an adaptation.

---

> ### Author Response · Authors · 2026-07-04
>
> We thank the reviewer for the constructive and precise suggestions. We address the two requested changes and the noted weaknesses below.
>
> **1. Isolating residual quantization from k-means initialization.** To separate the two levers we introduce, we added a paragraph to Section 5 that reads the decomposition directly off the audio results in Table 2, which already reports all three configurations along a single axis: single-codebook VQ, RQ with uniform initialization, and RQ with $k$-means. On AudioSet-2M linear probing at iteration 2, the total VQ→RQ($k$-means) improvement of +2.52 mAP separates cleanly into a quantizer effect of +1.67 (VQ→RQ, with initialization held data-agnostic) and an initialization effect of +0.85 (uniform→$k$-means, with the quantizer fixed); iteration 1 gives +2.39 and +0.43. Two conclusions follow. First, the residual quantizer, not the initialization, is the dominant lever — it accounts for roughly two-thirds to five-sixths of the linear-probing gain — with $k$-means a smaller but consistently positive addition. Second, because the effect appears at both bidirectional iterations, it is not an artifact of the training schedule. We are also explicit about a limitation: this clean three-way separation is available only for audio, since the image and video tables report VQ against RQ($k$-means) alone, so we restrict the quantitative attribution to audio and treat the other modalities as confirming only the combined direction.
>
> **2. Explicit algorithmic description and comparison to BEATs.** We added Algorithm 1, which states the full interleaved pretraining procedure and tags each step as either inherited unchanged from BEATs (**[B]**) or specific to BRIDLE (**[R]**). This speaks directly to the question of whether the interleaving is genuinely new or an adaptation: the interleaved encoder–tokenizer loop, the cold-start first iteration, the masked-prediction loss, and the EMA update are inherited, whereas the residual-quantization target, the $k$-means initialization, and the first-batch code reset are the changes. In other words, BRIDLE substitutes a hierarchical-residual tokenizer into the existing BEATs loop rather than proposing a new loop — the same positioning captured component-by-component in the new Table 1.
>
> **On the two weaknesses.** On the modest audio gains: fine-tuning at the base-ViT scale is close to saturated, which is why the improvements there are small; the more diagnostic signal is linear probing, where the VQ→RQ gain is larger (for example +2.52 mAP on AudioSet-2M) and, as the reviewer observes, points to stronger learned representations rather than fine-tuning effects. On novelty: we have deliberately positioned BRIDLE as a focused, controlled study of a single design choice — flat VQ versus hierarchical RQ targets — that transfers across three modalities, rather than as a new framework. We have also tempered the cross-modal claim: state-of-the-art is now claimed only for audio, external baselines have been added to the image and video tables (Tables 3–4), and those modalities are framed as evidence of cross-modality generality rather than per-modality superiority.
>
> We are grateful for suggestions that sharpened both what the paper claims and how clearly it relates to BEATs, and we hope the revision resolves the reviewer's concerns.